# On the Separability of Information in Diffusion Models

**Akhil Premkumar** [1]

## Abstract

Diffusion models transform noise into data by injecting information that was captured in their neural network during the training phase. In this paper, we ask: *what* is this information? We find that, in pixel-space diffusion models, (1) a large fraction of the total information in the neural network is committed to reconstructing small-scale perceptual details of the image, and (2) the correlations between images and their class labels are informed by the semantic content of the images, and are largely agnostic to the low-level details. We argue that these properties are intrinsically tied to the manifold structure of the data itself. Finally, we show that these facts explain the efficacy of classifier-free guidance: the guidance vector amplifies the mutual information between images and conditioning signals early in the generative process, influencing semantic structure, but tapers out as perceptual details are filled in.

## 1. Introduction

Conditional diffusion models must balance two objectives: they must learn to generate high-fidelity samples that capture the full complexity of the data distribution, including fine-grained structure and low-level details, while simultaneously learning the relationship between these samples and the conditioning information. Understanding how model capacity is allocated between these reconstruction and conditioning objectives is fundamental to diffusion model design. In this paper, we analyze this allocation in pixel-space diffusion models through the lens of information theory.

The goal of generative modeling is to produce samples of a random variable $\boldsymbol{X}$, given only training data drawn from its distribution $p_{\mathrm{d}}(\boldsymbol{x})$. Diffusion models accomplish this

by progressively denoising Gaussian random vectors till they are approximately distributed as $p_{\mathrm{d}}(\boldsymbol{x})$. To affect this transformation, the models reinstate information that was eroded by a forward diffusion process as it transformed the data into noise. This information, which is stored in a neural network, is quantified by the *neural entropy* $S_{\mathrm{NN}}^{\boldsymbol{x}}$ (Premkumar, 2025). It measures the effort required to collapse a Gaussian ball into $p_{\mathrm{d}}(\boldsymbol{x})$. A conditional model of $\boldsymbol{X}|\boldsymbol{Y}$ generates more targeted samples than an unconditional one since, on average, $p_{\mathrm{d}}(\boldsymbol{x}|\boldsymbol{y})$ is narrower than $p_{\mathrm{d}}(\boldsymbol{x})$. That is, conditioning with $\boldsymbol{Y}$ limits the range of possibilities for the value of $\boldsymbol{X}$. Therefore, a conditional model has to store more information since it must concentrate the Gaussian ball into even smaller distributions. Indeed, it can be shown that the neural entropy in a conditional model of $\boldsymbol{X}|\boldsymbol{Y}$ is $S_{\mathrm{NN}}^{\boldsymbol{x}|\boldsymbol{Y}} = S_{\mathrm{NN}}^{\boldsymbol{x}} + I(\boldsymbol{X};\boldsymbol{Y})$ (cf. Eq. (10)).

Real-world diffusion models are often applied to problems where the mutual information $I(\boldsymbol{X};\boldsymbol{Y})$ between the data variable $\boldsymbol{X}$ and the conditioning variable $\boldsymbol{Y}$ is low compared to the total information stored in the model. An unconditional model trained on $\boldsymbol{X}$ alone stores $S_{\mathrm{NN}}^{\boldsymbol{x}}$ nats of information, which is different from $S(\boldsymbol{X})$, the entropy of $\boldsymbol{X}$. Neural entropy is very large if $\boldsymbol{X}$ inhabits a lower-dimensional manifold relative to its naive coordinate representation, since a far greater reduction in uncertainty is required to locate such a manifold, starting from a Gaussian, than it would take if $\boldsymbol{X}$ fully occupied the ambient dimensions. A key point in this paper is that *the information required to resolve the $\boldsymbol{X}$ manifold precisely is largely irrelevant in correlating $\boldsymbol{X}$ with $\boldsymbol{Y}$*. This is because most of $S_{\mathrm{NN}}^{\boldsymbol{x}}$ is used up in transporting the probability mass *on to* the manifold, whereas $I(\boldsymbol{X};\boldsymbol{Y})$ locates $\boldsymbol{X}|\boldsymbol{Y}$ *within* the manifold (see Fig. 1).

At an operational level, the score function must become large at the later stages of the reverse process to squeeze the probability mass into a thin sliver of the total volume. This is what makes $S_{\mathrm{NN}}^{\boldsymbol{x}}$ large (cf. Eq. (11)). However, diffusion models can be used to reliably compute $I(\boldsymbol{X};\boldsymbol{Y})$ even in this situation, using a formula that involves the difference of the conditional and unconditional scores (Franzese et al., 2024; Kong et al., 2024). The divergent behavior in these scores is identical, and therefore, they cancel in the difference vector. This is the same as the 'classification vector' used to amplify the conditioning signal in classifier-free

---

[1]Department of Physics, University of California San Diego, La Jolla, CA 92093, USA. Work done while at Yale University. Correspondence to: <akhilprem.k@gmail.com>.

*Proceedings of the 43rd International Conference on Machine Learning*, Seoul, South Korea. PMLR 306, 2026. Copyright 2026 by the author(s).

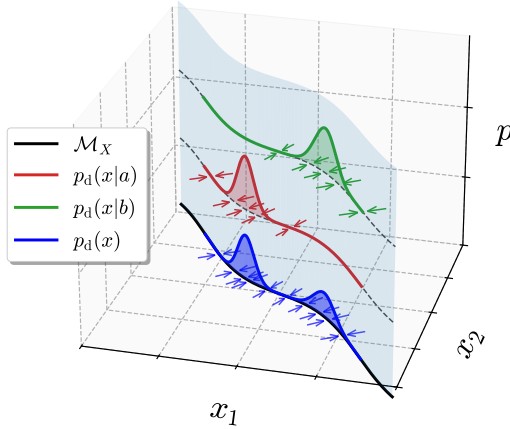 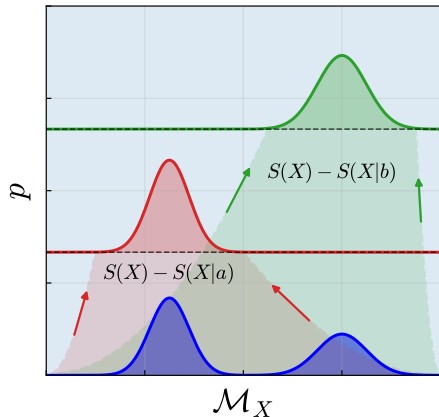

*Figure 1.* The 'orthogonality' of mutual information and (a large partion of) neural entropy, illustrated with a simple example. The thick black line, $\mathcal{M}_{\boldsymbol{X}}$, is a 1D manifold in the $\mathbb{R}^2$ plane. The marginal distribution $p_{\mathrm{d}}(\boldsymbol{x})$ lives on $\mathcal{M}_{\boldsymbol{X}}$, and it is a mixture of two Gaussian components labeled with $\boldsymbol{y} = a$ and $\boldsymbol{y} = b$. TC($\boldsymbol{X}$) is infinite, since $p_{\mathrm{d}}(\boldsymbol{x}) \equiv p(x_1, x_2)$ is supported on $\mathcal{M}_{\boldsymbol{X}}$ whereas $p_{\mathrm{d}}(x_1)p_{\mathrm{d}}(x_2)$ is supported on $\mathbb{R}^2$ (see Sec. 4). For this reason, the neural entropy required to convert $p_{\mathrm{eq}} \to p_{\mathrm{d}}$ also diverges (cf. Eq. (7)). The conditionals $p_{\mathrm{d}}(\boldsymbol{x}|\boldsymbol{y})$ are shown hovering above the marginal, with the bands (right) illustrating how specifying $\boldsymbol{y}$ reduces the entropy. The average reduction in entropy from $S(\boldsymbol{X}) \to S(\boldsymbol{X}|\boldsymbol{y})$ is the mutual information $I(\boldsymbol{X}; \boldsymbol{Y})$ (see Sec. A.1). The arrows (left) show the reverse drift vector confining the probability mass onto $\mathcal{M}_{\boldsymbol{X}}$ as $t \to T$ (cf. Eq. (2)). To accomplish this, the score functions become large in directions orthogonal to $\mathcal{M}_{\boldsymbol{X}}$, which is another way of seeing why $S_{\mathrm{NN}}$ is divergent. Also note that *both* conditional and unconditional scores become large at this instant, so their divergences cancel in the formula for $I(\boldsymbol{X}; \boldsymbol{Y})$ (cf. Eq. (12)). More intuitively, $S_{\mathrm{NN}}$ *is applied to collapse probability on to $\mathcal{M}_{\boldsymbol{X}}$ (left), whereas $I(\boldsymbol{X}; \boldsymbol{Y})$ is contracts the distribution along $\mathcal{M}_{\boldsymbol{X}}$ (right).*

guidance (CFG) (Ho & Salimans, 2022). Thus, the effectiveness of CFG is connected to the model's ability to estimate $I(\boldsymbol{X}; \boldsymbol{Y})$. We show that CFG increases $I(\boldsymbol{X}; \boldsymbol{Y})$, albeit at the expense of distorting the $\boldsymbol{X}$ distribution. Furthermore, the effect of CFG saturates beyond a certain value of the guidance strength parameter.

In image diffusion models, a substantial portion of $S_{\mathrm{NN}}^{\boldsymbol{x}}$ is used up in resolving the finer textures of images $\boldsymbol{X}$. Reproducing these details is tantamount to precisely locating the manifold in the final steps of the generative process (see Sec. 4). We show this by identifying within $S_{\mathrm{NN}}^{\boldsymbol{x}}$ a term called the *total correlation*, TC($\boldsymbol{X}$), which is a higher-dimensional generalization of mutual information (Watanabe, 1960). It measures the joint correlation between the pixels within $\boldsymbol{X}$. To get the textures right, the model must establish tight correlations between neighboring pixels, which incurs a huge information cost (see Fig. 1). Furthermore, these small-scale details are largely shared between different image classes and contribute little to distinguishing images with different macroscopic features. This is why $I(\boldsymbol{X}; \boldsymbol{Y})$ is low and $S_{\mathrm{NN}}^{\boldsymbol{x}}$ is large (see Fig. 9). These properties become manifest if we allow a diffusion model to generate its own conditioning signal $\boldsymbol{Z}$ by pairing it with an $\boldsymbol{X} \to \boldsymbol{Z}$ encoder. By varying the time intervals that modulate $\boldsymbol{Z}$, we show that diffusion models pick up little to no class-specific information from the textures, although they eat up a large fraction of the information budget.

**Contributions** In this paper we (1) identify within a conditional diffusion model the amount of information required to correlate the components within $\boldsymbol{X}$, and separately $\boldsymbol{X}$ with $\boldsymbol{Y}$, (2) show that the former overwhelms the latter in image diffusion models, $S_{\mathrm{NN}}^{\boldsymbol{x}} \gg I(\boldsymbol{X}; \boldsymbol{Y})$, (3) connect the manifold structure of images $\boldsymbol{X}$ to their perceptual and semantic content, (4) demonstrate that $I(\boldsymbol{X}; \boldsymbol{Y})$, which quantifies the correlation between images and their labels, is primarily sourced from the semantics.

## 2. Information Capture

Given a set of data vectors $\{\boldsymbol{x}^{(i)}\}_{i=1}^N$ in $\mathbb{R}^{D_{\boldsymbol{x}}}$, a probabilistic model approximates the underlying distribution $p_{\mathrm{d}}$ from which these vectors could have been sampled. One way to do this is to transform a generic initial distribution $p_0$ into one that is more likely to have produced the given samples. If $p_0$ is nearly the equilibrium state of a diffusive process (see Sec. F for notation)

$$\mathrm{d}\tilde{\boldsymbol{X}}_s = b_+(\tilde{\boldsymbol{X}}_s, s)\mathrm{d}s + \sigma(s)\mathrm{d}\hat{\boldsymbol{B}}_s, \tag{1}$$

then the transformation we seek is simply a reversal (playback) of the forward evolution that converts $p_{\mathrm{d}} \to p_0$ according to Eq. (1). The reverse process is effected by

$$\mathrm{d}\boldsymbol{X}_t = -(b_+(\boldsymbol{X}_t, T - t) - \sigma(T - t)^2 \nabla \log p(\boldsymbol{X}_t, t))\mathrm{d}t$$
$$+ \sigma(T - t)\mathrm{d}\boldsymbol{B}_t, \tag{2}$$

where $t := T - s$ is a time variable that runs in the opposite direction to $s$, and $p$ is the density that interpolates $p_0$ and $p_d$ (see Fig. 19). Diffusion is a dissipative process that erases information over time, which means reversal must reinstate the same amount of information to drive $p_0$ back to $p_d$. If $p_d$ is subject to Eq. (1) for a time $T$, and $b_+, \sigma$ have the same time-dependence, the information that must be injected to return to $p_d$ is quantified by the total entropy produced (Premkumar, 2025),

$$S_{\text{tot}} := \int_0^T \mathrm{d}t \, \frac{\sigma^2}{2} \mathbb{E}_p \left[ \|\nabla \log p_{\text{eq}} - \nabla \log p\|^2 \right]$$
$$= D_{\text{KL}} \left( p_d \| p_{\text{eq}} \right) - D_{\text{KL}} \left( p_0 \| p_{\text{eq}} \right). \quad (3)$$

The expectation is taken over trajectories generated by Eq. (1), starting at $\tilde{\boldsymbol{X}}_0 \sim p_d$, and $p_{\text{eq}}$ is the quasi-invariant state, which can be understood as the 'least informative state' consistent with the forward dynamics. It is the distribution that would result if we froze $b_+$ and $\sigma$ at their values at $t$ and waited for the system to equilibrate. That is, $p_{\text{eq}}(x) \propto \exp \left[ \int^x 2b_+/\sigma^2 \right]$. In a diffusion model, the drift term in Eq. (2) is approximated by a neural network. It is useful to parameterize the reverse SDE as

$$\mathrm{d}\boldsymbol{X}_t = (b_+(\boldsymbol{X}_t, T - t) + \sigma(T - t)^2 \boldsymbol{e_\theta}(\boldsymbol{X}_t, T - t))\mathrm{d}t$$
$$+ \sigma(T - t)\mathrm{d}\boldsymbol{B}_t, \quad (4)$$

where the network $\boldsymbol{e_\theta}$ is trained to minimize (cf. Sec. B.3)

$$\mathcal{L}_{\text{EM}} = \int_0^T \mathrm{d}t \, \frac{\sigma^2}{2} \mathbb{E}_p \left[ \|\nabla \log p_{\text{eq}} - \nabla \log p + \boldsymbol{e_\theta}\|^2 \right]. \quad (5)$$

If $\boldsymbol{e_\theta} = 0$, Eq. (4) reduces to the forward dynamics, Eq. (1). Let $\mathcal{P}[p_d]$ be the probability that $N$ random vectors from $p_0$ would be distributed as $p_d$ at $t = T$ under Eq. (1). A *perfectly* trained diffusion model, with the idealized network $\boldsymbol{e_\theta^\star} = -2b_+/\sigma^2 + \nabla \log p$, modifies the dynamics to Eq. (2), which is guaranteed to take $p_0 \to p_d$. Such a network stores/applies precisely $S_{\text{tot}}$ worth of information to affect this transformation, since $S_{\text{tot}} = -\frac{1}{N} \log \mathcal{P}[p_d]$. This is why they are called *entropy-matching* models.

Information negates uncertainty. The idealized entropy-matching model applies $S_{\text{tot}}$ worth of information to reconstitute $p_d$ from $p_0$ in time $T$. If $T$ is large enough that $p_0 \approx p_{\text{eq}}$, the total entropy can be written as

$$S_{\text{tot}}^{\boldsymbol{X}} = D_{\text{KL}} \left( p_d(\boldsymbol{x}) \| p_{\text{eq}}(\boldsymbol{x}) \right)$$
$$= -S(\boldsymbol{X}) - \int \mathrm{d}\boldsymbol{x} \, p_d(\boldsymbol{x}) \log p_{\text{eq}}(\boldsymbol{x}), \quad (6)$$

We have introduced a superscript $\boldsymbol{X}$ in $S_{\text{tot}}^{\boldsymbol{x}}$ to specify explicitly the random variable whose distribution is being modeled. Crucially, $S_{\text{tot}}^{\boldsymbol{x}}$ is *not* the same as $S(\boldsymbol{X})$. In fact, $S_{\text{tot}}^{\boldsymbol{x}}$ is larger if $S(\boldsymbol{X})$ is smaller, since the diffusion model must apply more information to locate a narrower $p_d(\boldsymbol{x})$.

This is especially a problem in continuous diffusion models, where $S(\boldsymbol{X})$ can be arbitrarily small; the differential entropy of a Dirac delta function is $-\infty$. Such divergences arise when $\boldsymbol{X}$ lives in a lower-dimensional manifold, as we explore further in Sec. 4.

Eq. (6) also reveals an interesting fact about correlations within the components of $\boldsymbol{X}$. In the unconditional case, with $T$ large enough that $p_0 \approx p_{\text{eq}}$, the total entropy can be factorized as

$$S_{\text{tot}} = D_{\text{KL}} \left( p_d \| p_{\text{eq}} \right)$$
$$= \sum_{k=1}^{D_{\boldsymbol{x}}} D_{\text{KL}} \left( p_d(x_k) \| p_{\text{eq}}(x_k) \right)$$
$$+ \underbrace{D_{\text{KL}} \left( p_d(x_1, \ldots, x_{D_{\boldsymbol{X}}}) \,\middle\|\, \prod_{k=1}^{D_{\boldsymbol{X}}} p_d(x_k) \right)}_{\text{TC}(\boldsymbol{X})}. \quad (7)$$

The last term, called the *total correlation*, is a generalization of mutual information to multiple random variables (Watanabe, 1960). In Eq. (7) $x_k$ is the $k$-th component of a data vector $\boldsymbol{x}$, and $p_d(x_k)$ and $p_{\text{eq}}(x_k)$ are the marginal densities obtained by integrating $p_d(\boldsymbol{x})$ and $p_{\text{eq}}(\boldsymbol{x})$ over all components except $x_k$. Eq. (7) tells us that during the reversal/generative stage, the model must (1) shift the marginals for each $x_k$ from $p_{\text{eq}}(x_k) \to p_d(x_k)$, and (2) establish correlations between different $x_k$. Thus, denoising a vector from $p_0$ is, in part, the process of *restoring the component-wise correlations* that were lost in the forward stage.

Next, we consider a scenario where the diffusion model is used for conditional generation. Let $\boldsymbol{Y}$ be the conditioning information. For example, $\boldsymbol{Y}$ represents the class labels in class-conditioned image generation, with $\boldsymbol{X}$ being the associated images. Given $\boldsymbol{Y} = \boldsymbol{y}$, a new sample can be generated by applying Eq. (4) with

$$\boldsymbol{e_\theta^\star}(\boldsymbol{x}_t, T - t; \boldsymbol{y}) = -\frac{2b_+(\boldsymbol{x}_t, T - t)}{\sigma^2(T - t)} + \nabla \log p(\boldsymbol{x}_t, t | \boldsymbol{y}). \quad (8)$$

Let $S_{\text{tot}}^{\boldsymbol{X}|\boldsymbol{y}}$ denote the information stored by such a network for each $\boldsymbol{y}$. On average, this model injects an amount of information

$$S_{\text{tot}}^{\boldsymbol{X}|\boldsymbol{Y}} := \mathbb{E}_{\boldsymbol{Y}} \left[ S_{\text{tot}}^{\boldsymbol{X}|\boldsymbol{y}} \right] = \mathbb{E}_{\boldsymbol{Y}} \left[ D_{\text{KL}} \left( p_d(\boldsymbol{x}|\boldsymbol{y}) \| p_{\text{eq}}(\boldsymbol{x}) \right) \right]$$
$$= -S(\boldsymbol{X}|\boldsymbol{Y}) - \int \mathrm{d}\boldsymbol{x} \, p_d(\boldsymbol{x}) \log p_{\text{eq}}(\boldsymbol{x}). \quad (9)$$

We expect $S_{\text{tot}}^{\boldsymbol{X}|\boldsymbol{Y}} \geq S_{\text{tot}}^{\boldsymbol{x}}$, since more information is needed to squeeze the quasi-invariant state into the distributions $p_d(\boldsymbol{x}|\boldsymbol{y})$, which are on average narrower than the marginal $p_d(\boldsymbol{x})$ (see Sec. A.1). Indeed, the conditional model injects an *additional* amount of information

$$S_{\text{tot}}^{\boldsymbol{X}|\boldsymbol{Y}} - S_{\text{tot}}^{\boldsymbol{X}} = S(\boldsymbol{X}) - S(\boldsymbol{X}|\boldsymbol{Y}) \equiv I(\boldsymbol{X}; \boldsymbol{Y}). \quad (10)$$

Importantly, this $I(\boldsymbol{X}; \boldsymbol{Y})$ amount of information is stored atop $S_{\text{tot}}^{\boldsymbol{X}}$, which is different from $S(\boldsymbol{X})$. The diffusion model stores/injects $I(\boldsymbol{X}; \boldsymbol{Y})$ nats of extra information, which it uses to correlate $\boldsymbol{X}$ to $\boldsymbol{Y}$. In a similar way, the $\text{TC}(\boldsymbol{X})$ piece of $S_{\text{tot}}^{\boldsymbol{X}}$ is used to correlate the internal components of $\boldsymbol{X}$ with one another jointly.

**Entropy-matching**  In the discussion above, we have parameterized the reverse drift in Eq. (4) as $b_+ + \sigma^2 \boldsymbol{e_\theta}$, which is different from the score-matching parameterization of $-b_+ + \sigma^2 \boldsymbol{s_\theta}$. The latter forces the network to retain additional information to counteract the repulsive $-b_+$ term, as explained in (Premkumar, 2025). A simple thought experiment reveals the problem: suppose we take $p_{\text{d}} = p_0 \approx p_{\text{eq}}$. The forward process has little effect on the distribution since $p_{\text{d}}$ is already close to equilibrium. However, the scores for this transformation are still non-zero over the support of $p_{\text{eq}}$, which means the network in a score-matching model must retain information to convert a distribution *back to itself*. On the other hand, an entropy-matching network would store no information in this scenario, as expected. In this sense, entropy-matching makes transparent the correspondence between the network's information content and the entropy of the underlying data. We can convert a score-matching model to an entropy matching one with the simple substitution $\boldsymbol{s_\theta} = \nabla \log p_{\text{eq}} + \boldsymbol{e_\theta}$.

**Neural Entropy**  We derived Eq. (10) under the assumption of an ideal entropy-matching model $\boldsymbol{e_\theta^\star}$, which absorbs exactly $S_{\text{tot}}$ units of information during training. In practice, no model achieves this ideal because of the finite number of training epochs, limited batch size, and finite data. However, (Premkumar, 2025) demonstrates that the amount of information stored in a real network $\boldsymbol{e_\theta}$ is measured through its *neural entropy*,

$$S_{\text{NN}}^{\boldsymbol{X}} := \int_0^T ds \, \frac{\sigma(s)^2}{2} \mathbb{E}_p \left[ \|\boldsymbol{e_\theta}(\tilde{\boldsymbol{x}}_s, s)\|^2 \right] \approx S_{\text{tot}}^{\boldsymbol{X}}. \quad (11)$$

Notice that setting $\boldsymbol{e_\theta} \to \boldsymbol{e_\theta^\star}$ turns $S_{\text{NN}} \to S_{\text{tot}}$, which follows from Eqs. (3) and (5). Away from this theoretical limit, the neural entropy can be either smaller or larger than the true $S_{\text{tot}}$. For example, when the dataset is sparse, the diffusion model tends to concentrate probability mass around the available samples, demanding greater effort from the network than it requires to reconstitute the true $p_{\text{d}}$, which may have a more distributed support. Another possibility is that training is not long enough for the network to absorb all of $S_{\text{tot}}$, so neural entropy trails the true value. Nevertheless, with sufficient training and a large enough dataset, Eq. (11) provides a close approximation to $S_{\text{tot}}$ (see Fig. 12). These are the entropy-matching models we discuss henceforth.

## 3. Mutual Information and Guidance

It is possible to use Eq. (10) to estimate the mutual information between two high-dimensional random variables. Given a set of pairs $\{(\boldsymbol{x}^{(i)}, \boldsymbol{y}^{(i)})\}_{i=1}^N$ we can train an entropy-matching model to reconstruct the distribution of $\boldsymbol{X}$ given $\boldsymbol{Y}$ and, separately, the marginal distribution of $\boldsymbol{X}$. The difference in neural entropy between the two approximates $I(\boldsymbol{X}; \boldsymbol{Y})$. This approach is closely related to the results in (Franzese et al., 2024; Kong et al., 2024). Specifically, Eq. (19) in the former says

$$I(\boldsymbol{X}; \boldsymbol{Y})$$
$$= \mathbb{E}_{\boldsymbol{Y}} \left[ \int_0^T ds \, \frac{\sigma^2}{2} \mathbb{E}_{\tilde{\boldsymbol{X}}_s, \boldsymbol{X}|\boldsymbol{y}} \left\| \begin{array}{c} \nabla \log p(\tilde{\boldsymbol{x}}_s, s|\boldsymbol{y}) \\ - \nabla \log p(\tilde{\boldsymbol{x}}_s, s) \end{array} \right\|^2 \right]$$
$$(12)$$
$$\approx \mathbb{E}_{\boldsymbol{Y}} \left[ \int_0^T ds \, \frac{\sigma^2}{2} \mathbb{E}_{\tilde{\boldsymbol{X}}_s, \boldsymbol{X}|\boldsymbol{y}} \|\boldsymbol{e_\theta}(\tilde{\boldsymbol{x}}_s, s; \boldsymbol{y}) - \boldsymbol{e_\theta}(\tilde{\boldsymbol{x}}_s, s)\|^2 \right]$$
$$(13)$$

up to terms that vanish as $T \to 0$, and when the $\boldsymbol{e_\theta}$'s are a good approximation of their true values $\boldsymbol{e_\theta^\star}$. We give an alternative derivation of this result in Sec. C.1. Notice that Eq. (13) also works with score-matching models, with $\boldsymbol{e_\theta}$ replaced by the corresponding $\boldsymbol{s_\theta}$. The main advantage of entropy-matching is that it links the information stored in the network to the effort required to reconstitute $p_{\text{d}}$. For now, we consider how Eq. (12) helps us better understand classifier-free guidance (CFG) (Ho & Salimans, 2022).

In image models, $\boldsymbol{X}$ denotes images and $\boldsymbol{Y}$ the corresponding class labels. (Dhariwal & Nichol, 2021) show that the quality of generated samples can be improved, at the cost of decreased diversity, by forcing the model to adhere more strongly to the conditioning variable. If Eq. (2) evolves $p_0$ to $p(\boldsymbol{x}_t, t|\boldsymbol{y})$ at time $t$, the conditioning on $\boldsymbol{y}$ can be amplified by modifying the drift vectors to sample from $p(\boldsymbol{x}_t, t|\boldsymbol{y}) p(\boldsymbol{y}|\boldsymbol{x}_t, t)^w$ instead, where $w > 0$ is a parameter we can control. (Ho & Salimans, 2022) accomplish this by constructing an implicit classifier $p(\boldsymbol{y}|\boldsymbol{x}_t, t) \propto p(\boldsymbol{x}_t, t|\boldsymbol{y})/p(\boldsymbol{x}_t, t)$, which has the score

$$\boldsymbol{s}_{\text{cl}}(\boldsymbol{y}, t) := \nabla_{\boldsymbol{x}_t} \log p(\boldsymbol{y}|\boldsymbol{x}_t, t)$$
$$= \nabla_{\boldsymbol{x}_t} \log p(\boldsymbol{x}_t, t|\boldsymbol{y}) - \nabla_{\boldsymbol{x}_t} \log p(\boldsymbol{x}_t, t)$$
$$\equiv \nabla_{\tilde{\boldsymbol{x}}_s} \log p(\tilde{\boldsymbol{x}}_s, s|\boldsymbol{y}) - \nabla_{\tilde{\boldsymbol{x}}_s} \log p(\tilde{\boldsymbol{x}}_s, s)$$
$$\approx \boldsymbol{e_\theta}(\tilde{\boldsymbol{x}}_s, s; \boldsymbol{y}) - \boldsymbol{e_\theta}(\tilde{\boldsymbol{x}}_s, s). \quad (14)$$

This is also the vector whose $\ell_2$-norm appears in Eq. (12). A close examination of the latter helps us build some intuition for $\boldsymbol{s}_{\text{cl}}$. We begin by noting that the inner expectation in Eq. (12) is computed over trajectories that start from samples of $\boldsymbol{X}|\boldsymbol{Y} = \boldsymbol{y}$ (or $\boldsymbol{X}|\boldsymbol{y}$ for short). If we were to average $\|\boldsymbol{s}_{\text{cl}}(\boldsymbol{y}, s)\|^2$ over forward paths that emanate from

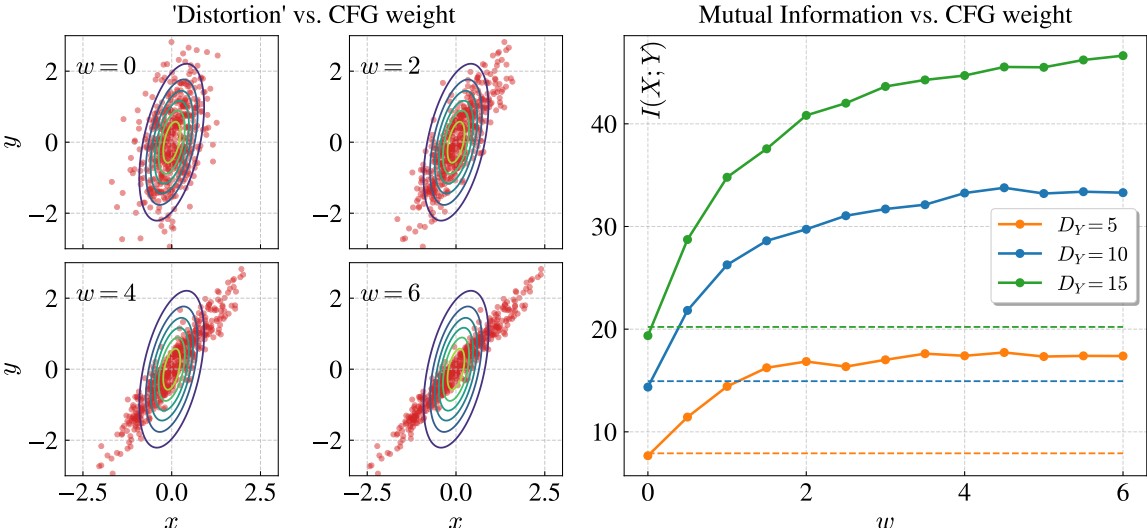

*Figure 2.* **Left:** Samples generated by a CFG-style modification to the conditional score $\nabla \log p(x_t, t|y)$ of a joint Gaussian, $\mathbf{Y} = A\mathbf{X} + \boldsymbol{\varepsilon}$ (cf. Eqs. (15) and (47)). CFG strengthens the correlation between $\mathbf{X}$ and $\mathbf{Y}$, increasing their mutual information. But it also alters the relationship between them. **Right:** Mutual information under CFG for the joint Gaussian. We fix $D_{\mathbf{X}} = 25$ and repeat the experiment with $D_{\mathbf{Y}} = 5, 10, 15$. Notice how $I(\mathbf{X}; \mathbf{Y})_{\text{CFG}}$ increases as the guidance strength is ramped up (dashed lines are the original $I(\mathbf{X}; \mathbf{Y})$ values, cf. Eq. (49)). It saturates faster for smaller $D_{\mathbf{Y}}$, when $\mathbf{Y}$ has fewer degrees of freedom to encode the diversity in $\mathbf{X}$. See Figs. 8 and 10 for more details.

$\mathbf{X}|\mathbf{y}'$ where $\mathbf{y}' \neq \mathbf{y}$, the expectation value would be higher, since the conditional score $\nabla \log p(\tilde{\mathbf{x}}_s, s|\mathbf{y})$ along those paths are larger. That is, the conditional scores are steeper in regions explored by diffused versions of $\mathbf{X}|\mathbf{y}'$, where the underlying conditional density $p(\tilde{\mathbf{x}}_s, s|\mathbf{y})$ falls off rapidly. Therefore, the integral over $\mathbb{E}_{\tilde{\mathbf{X}}_s, \mathbf{X}|\mathbf{y}'} \|\mathbf{s}_{\text{cl}}(\mathbf{y}, s)\|^2$ is *smallest* when $\mathbf{y}' = \mathbf{y}$.[1] During the generative process, a sample can be made more $\mathbf{y}$-like by augmenting the reverse drift term in Eq. (2) with $w \times \mathbf{s}_{\text{cl}}(\mathbf{y}, t)$. That is, we replace $\nabla \log p(\mathbf{x}_t, t|\mathbf{y})$ with

$$(1 + w)\nabla \log p(\mathbf{x}_t, t|\mathbf{y}) - w\nabla_{\mathbf{x}_t} \log p(\mathbf{x}_t, t). \quad (15)$$

This is CFG. The new drift term pulls a reverse-evolving $\mathbf{x}_t$ more strongly toward regions in $\mathbf{x}$-space that are consistent with the condition $\mathbf{y}$. At the next time step, $\mathbf{x}_{t+\Delta t}$ is more $\mathbf{y}$-like, so $\mathbf{s}_{\text{cl}}(\mathbf{y}, t)$ *weakens*. Therefore, the effect of CFG diminishes over time $t$. In the limit, the fully denoised sample will be more tightly determined by $\mathbf{y}$, so $I(\mathbf{X}; \mathbf{Y})_{\text{CFG}}$ will be higher. There is a limit, however, since the increase in $I(\mathbf{X}; \mathbf{Y})_{\text{CFG}}$ saturates at higher values of $w$ if the dimensionality of $\mathbf{Y}$ is smaller than that of $\mathbf{X}$ (see Fig. 2). In that case, $\mathbf{Y}$ lacks the sufficient code length to encode the information content of $\mathbf{X}$ faithfully; this is an example of an *information bottleneck* (see Sec. A.2).

One might wonder whether it is possible to substitute Eq. (15) into Eq. (12) to conclude that mutual informa-

tion is boosted to $(1 + w)^2 I(\mathbf{X}; \mathbf{Y})$. But that would be incorrect; such a maneuver is disallowed by the fact that the modified score does not correspond to any known forward diffusion process. For the same reason, the distribution we reconstruct under CFG is not the true $p_{\text{d}}$ (see Fig. 2). We discuss this point further in Sec. C.2. It remains true, however, that CFG strengthens the binding between $\mathbf{X}$ and $\mathbf{Y}$.

## 4. Manifolds and the Information Budget

Mutual information is largest if knowledge of the value of one variable completely determines the value of the other (see Sec. A.1). On the other hand, if a given value of $\mathbf{Y}$ corresponds to a wide range of $\mathbf{X}$, a greater amount of uncertainty remains about the value of the latter, so $I(\mathbf{X}; \mathbf{Y})$ is low. This is the case with labeled image datasets, where a label $\mathbf{Y} = \mathbf{y}$ can correspond to a rich distribution of images $\mathbf{X}|\mathbf{Y} = \mathbf{y}$. For instance, the label 'dog' corresponds to a wide variety of dogs. There is, however, a subtle point here about image datasets: the entropy of images does not arise only from high-level semantic variation (different breeds, poses, or scenes), but is overwhelmingly dominated by the low-level perceptual details present in each image. Diffusion models capture these details with remarkable fidelity, and a large share of their information capacity is devoted to encoding fine perceptual structure rather than high-level semantics. In fact, much of the low-level detail is not class-specific, but is shared across multiple categories of images (see Fig. 17). Therefore, specifying the label 'dog' does

---

[1] In fact, it is possible to build a classifier based on this very insight (Clark & Jaini, 2023; Li et al., 2023).

very little to narrow down the possibilities of which sample to draw. This is why diffusion models sometimes stray from the conditioning signal during generation: the mutual information between images and labels is intrinsically low compared to its neural entropy. We provide empirical proof of these statements in Sec. 5. For now, we will rationalize them from an information theory perspective.

We have made two assertions thus far: (1) neural entropy is high when $\boldsymbol{X}$ lives in a low-dimensional manifold, and (2) a sizable fraction of this information is used up in resolving minute perceptual details. These two points are closely related. First, the largeness of $S_{\mathrm{NN}}^{\boldsymbol{X}} \approx D_{\mathrm{KL}}(p_{\mathrm{d}} \| p_{\mathrm{eq}})$ is related to the fact that $p_{\mathrm{eq}}$ is supported on the entire volume, whereas $p_{\mathrm{d}}$ is restricted to the manifold, call it $\mathcal{M}_{\boldsymbol{X}}$. We can drill down further and triangulate the problem to the $\mathrm{TC}(\boldsymbol{X})$ from Eq. (7). The joint density $p_{\mathrm{d}}(x_1, \ldots, x_{D_{\boldsymbol{X}}})$ is supported on $\mathcal{M}_{\boldsymbol{X}}$ whereas the product density $\prod_{k=1}^{D_{\boldsymbol{X}}} p_{\mathrm{d}}(x_k)$ is supported over a larger joint space. Therefore, the KL between them diverges.

The same phenomenon can also occur in $I(X_1; X_2) = D_{\mathrm{KL}}(p(x_1, x_2) \| p(x_1)p(x_2))$, where $X_1$ and $X_2$ are two 1-D continuous random variables. Unlike the discrete case, $I(X_1; X_2)$ can be infinite if $X_1$ and $X_2$ are perfectly correlated—specifying a real number requires *infinitely many digits of precision*, so we gain an infinite amount of information about $X_1$ from a given $X_2 = x_2$. More formally, the joint density $p(x_1, x_2)$ is confined to a lower-dimensional manifold, since $X_2 = f(X_1)$, whereas the product density $p(x_1)p(x_2)$ is supported over the full joint space (see Fig. 1 and Sec. A.1).

Extending the same intuition to $\mathrm{TC}(\boldsymbol{X})$, we see that if the value of a pixel $x_k$ resolves the intensity of any other pixel to very high precision, $\mathrm{TC}(\boldsymbol{X})$ diverges. But this is what textures are: nearby pixels that are strongly correlated to one another. Under forward diffusion, these correlations are the first to vanish, since the less significant digits of every $x_k$ are overwhelmed faster. In simpler terms, with a little noise added, we can still make out the semantic details in the image, but the smaller-scale details would have washed away (see Sec. E.3). It follows from this argument that $S_{\mathrm{NN}}^{\boldsymbol{X}}$ should receive a large contribution from the neighborhood of $s = 0$ (or $t = T$). This is exactly what we observe empirically (see Fig. 3). The reverse drift term, and in particular $\nabla \log p(\boldsymbol{x}_t, t)$, becomes very large as $t \to T$ to confine the probability mass to the image manifold $\mathcal{M}_{\boldsymbol{X}}$. As we show in Sec. 5, $\nabla \log p(\boldsymbol{x}_t, t | \boldsymbol{y})$ also exhibits the same divergence, which cancels out the one from the marginal score in Eq. (12). So $I(\boldsymbol{X}; \boldsymbol{Y})$ receives little contribution from the final time steps, which resolve small-scale details. In other words, $\boldsymbol{Y}$ informs the semantic structure of the images at early $t$, leaving the model to fill in the perceptual detail later (see Fig. 1).

## 5. Experiments

**Joint Gaussian** Before delving into image models, it is worth considering a toy problem which allows precise control over $I(\boldsymbol{X}; \boldsymbol{Y})$ vis-à-vis $S_{\mathrm{NN}}^{\boldsymbol{x}}$. Consider the random variables $\boldsymbol{X} \sim \mathcal{N}(0, \Sigma_{\boldsymbol{X}})$, and $\boldsymbol{Y} = A\boldsymbol{X} + \boldsymbol{\varepsilon}$ (see Sec. D.1). From Eqs. (49) and (50), we have the closed form expressions:

$$S_{\mathrm{tot}}^{\boldsymbol{X}} \approx D_{\mathrm{KL}}(p_{\mathrm{d}} \| p_{\mathrm{eq}}) = \frac{1}{2} \left( \mathrm{Tr}(\Sigma_{\boldsymbol{X}}) - \log |\Sigma_{\boldsymbol{X}}| - D_{\boldsymbol{X}} \right),$$

$$I(\boldsymbol{X}; \boldsymbol{Y}) = \frac{1}{2} \log \left( \frac{|\Sigma_{\boldsymbol{Y}}|}{|\Sigma_{\boldsymbol{\varepsilon}}|} \right),$$

where we have assumed a VP forward process, for which $p_{\mathrm{eq}} = \mathcal{N}(0, I)$. The scores for this distribution are also known analytically, which allows us to compare the learned $e_{\boldsymbol{\theta}}$ vectors with their idealized versions $e_{\boldsymbol{\theta}}^{\star}$ (cf. Eq. (8)). In particular, we choose

$$\Sigma_{\boldsymbol{X}} = U\Lambda U^{\top} + \epsilon I, \quad \Lambda = \mathrm{diag}(\lambda_1, \cdots, \lambda_k, \overbrace{\lambda_\delta, \cdots, \lambda_\delta}^{D_{\boldsymbol{X}} - k}).$$

where $\Sigma_{\varepsilon} = \sigma_{\boldsymbol{\varepsilon}}^2 I$, $\lambda_{\bullet} > 0$, $U$ is an orthogonal matrix, and a small $\epsilon > 0$ ensures stability. This simple model allows us to study how well diffusion models absorb different amounts of $S_{\mathrm{tot}}^{\boldsymbol{x}}$ and $I(\boldsymbol{X}; \boldsymbol{Y})$. We do two kinds of experiments with this, corresponding to two different ways of making $S_{\mathrm{tot}}^{\boldsymbol{x} | \boldsymbol{Y}}$ large. (1) *Flattening*: We adjust $\lambda_\delta$ to be much smaller than the other eigenvalues to induce an approximately low-rank structure in $\Sigma_{\boldsymbol{X}}$. This 'flattens' $\boldsymbol{X}$ along $D_{\boldsymbol{X}} - k$ directions, which causes $S_{\mathrm{tot}}^{\boldsymbol{x}}$ to blow up, simulating the manifold problem. At the same time we maintain $\Sigma_{\boldsymbol{Y}} = A\Sigma_{\boldsymbol{X}}A^{\top} + \Sigma_{\boldsymbol{\varepsilon}}$ at full-rank by choosing $\Sigma_{\boldsymbol{\varepsilon}}$ appropriately, which keeps $I(\boldsymbol{X}; \boldsymbol{Y})$ under control (see Fig. 9). (2) *Determinism:* We keep $\Sigma_{\boldsymbol{X}}$ full rank and make $\boldsymbol{Y}$ a more deterministic function of $\boldsymbol{X}$ by making $\Sigma_{\boldsymbol{\varepsilon}}$ low (cf. Eq. (51)). Here, $S_{\mathrm{tot}}^{\boldsymbol{X}}$ remains constant, whereas $I(\boldsymbol{X}; \boldsymbol{Y})$ diverges. The results are shown in Fig. 4.

We train two diffusion models, one on samples of $\boldsymbol{X}$ alone, and the other on a joint sample of $(\boldsymbol{X}, \boldsymbol{Y})$. In the flattening experiment, both models struggle to absorb the respective $S_{\mathrm{tot}}$'s, especially from early $s$, but they manage to estimate $I(\boldsymbol{X}; \boldsymbol{Y})$ correctly! This is because the divergences in the conditional and unconditional scores cancel in Eq. (12), as evident from the green lines in the leftmost plot of Fig. 4. More intuitively, *the information required to resolve the manifold $\mathcal{M}_{\boldsymbol{X}}$ is largely irrelevant in correlating $\boldsymbol{X}$ with $\boldsymbol{Y}$*. On the other hand, if $\sigma_{\boldsymbol{\varepsilon}}$ is made small $\boldsymbol{X}$ and $\boldsymbol{Y}$ converge on the hyperplane $\boldsymbol{y} = A\boldsymbol{x}$. $\Sigma_{\boldsymbol{X}}$ is full rank, which keeps $S_{\mathrm{tot}}^{\boldsymbol{x}}$ under control. The conditional and unconditional scores no longer cancel out, since the former grows large to confine the probability mass to the hyperplane in the joint $\boldsymbol{xy}$ space, while the latter remains mild in comparison.

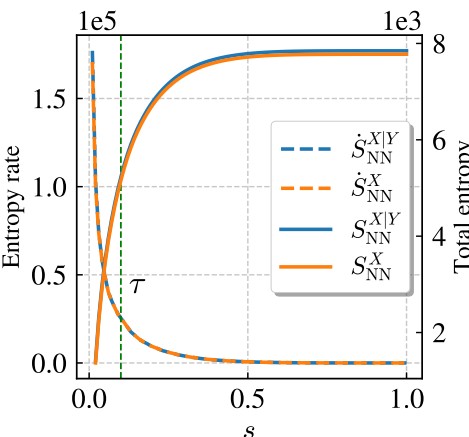

*Figure 3.* The neural entropy curves for CIFAR-10. Their time derivatives, called entropy rates, quantify the information erased/injected per unit time. The sharp rise in the rates as $s \to 0$ is due to the manifold structure of $\mathcal{M}_{\boldsymbol{X}}$. See Fig. 16 for more details, and experiments with MNIST and Tiny ImageNet.

The flattening experiment is a simplified version of what happens in image models. The ratio of $I(\boldsymbol{X}; \boldsymbol{Y})$ to $S_{\mathrm{tot}}^{\boldsymbol{x}}$ is much more extreme in the latter, hovering around $10^{-4}$ to $10^{-3}$ (see Fig. 16). Indeed, even in the Gaussian case, we observe that estimates of $I(\boldsymbol{X}; \boldsymbol{Y})$ deteriorate when $\lambda_\delta$ is made too small (see Fig. 13). Next, we study whether such coupling between $I(\boldsymbol{X}; \boldsymbol{Y})$ and the neural entropy can be detected in image models.

**Images**  We do not have access to the true scores for image distributions, so we cannot gauge whether $S_{\mathrm{NN}}$ and $I(\boldsymbol{X}; \boldsymbol{Y})$ from the network are close to their actual values (see Fig. 16). Therefore, we take an alternate route to ascertain the claims in Sec. 4. Specifically, we want to assess whether the early time steps in $s$, those that pinpoint the manifold and resolve the minute details, contain any class-specific information. To do this, we allow the diffusion model to produce its own conditioning information by pairing it with an encoder $q_\phi(\boldsymbol{z}|\boldsymbol{x})$. The resulting arrangement, called a diffusion autoencoder (DAE), is discussed in great detail in Sec. E.1. We make some modifications.

Briefly, the DAE produces a compressed representation $\boldsymbol{Z}$ of the images $\boldsymbol{X}$. The trick here is to note that we can condition a diffusion model on different $\boldsymbol{Z}$'s at different stages of the diffusion process. We use two latents, $\boldsymbol{Z}_{\mathrm{per}}$ and $\boldsymbol{Z}_{\mathrm{sem}}$, and split the diffusion model loss into (see Sec. E.3 for details)

$$\mathcal{L}_{\mathrm{DEM}}^{\mathrm{split}} := \int_0^\tau \mathrm{d}s \, \mathbb{E}_{\boldsymbol{Z}_{\mathrm{per}}}[L(\boldsymbol{z}_{\mathrm{per}}; s)]$$
$$+ \int_\tau^T \mathrm{d}s \, \mathbb{E}_{\boldsymbol{Z}_{\mathrm{sem}}}[L(\boldsymbol{z}_{\mathrm{sem}}; s)], \quad (16)$$

$$L(\boldsymbol{z}; s) := \mathbb{E}_{\boldsymbol{X}, \tilde{\boldsymbol{x}}_s}\left[\frac{\sigma^2}{2}\left\|\nabla \log p_{\mathrm{eq}}^{(s)}(\tilde{\boldsymbol{x}}_s)\right. \quad (17)$$
$$\left.-\nabla \log p(\tilde{\boldsymbol{x}}_s, s|\boldsymbol{x}, 0) + \boldsymbol{e}_{\boldsymbol{\theta}}(\tilde{\boldsymbol{x}}_s, s; \boldsymbol{z})\right\|^2\right].$$

The resulting arrangement encodes into $\boldsymbol{Z}_{\mathrm{per}}$ only the information that was absorbed from $s \in (0, \tau)$ by the diffusion model. The remaining information, from $s \in [\tau, T)$, imprints on $\boldsymbol{Z}_{\mathrm{sem}}$. Thus, $\boldsymbol{Z}_{\mathrm{per}}$ and $\boldsymbol{Z}_{\mathrm{sem}}$ are compressed proxies for what the diffusion model 'sees' in those intervals (see Fig. 3). The DAE creates these latents from $\boldsymbol{X}$ alone; it is not given the class labels $\boldsymbol{Y}$. Therefore, we can check how well $\boldsymbol{Z}_{\mathrm{per}}$ and $\boldsymbol{Z}_{\mathrm{sem}}$ correlate with $\boldsymbol{Y}$ to understand which stages of diffusion are informed by $\boldsymbol{Y}$. This is done by estimating $I(\boldsymbol{Z}_{\mathrm{per}}; \boldsymbol{Y})$ and $I(\boldsymbol{Z}_{\mathrm{sem}}; \boldsymbol{Y})$—the distributions of $\boldsymbol{Z}_\bullet$ are not low-rank, so we train conditional diffusion models on $\boldsymbol{Z}_\bullet|\boldsymbol{Y}$ and use Eq. (12). Next, we scan $\tau$ from $0.1T \to 0.9T$, training a new DAE at each $\tau$. The results are shown in Fig. 5, and Figs. 17 and 18.

A t-SNE diagram of the two latents immediately confirms our thesis: $\boldsymbol{Z}_{\mathrm{per}}$ shows little structure that reflects any class-specific clustering at $\tau = 0.1T$, indicating that there is little $\boldsymbol{Y}$-dependent information at early $s$. On the other hand, $\boldsymbol{Z}_{\mathrm{sem}}$ shows clear separation of clusters—the model can identify the digits from slightly noisy images of them. Furthermore, as we increase $\tau$ more $\boldsymbol{Y}$-specific information leaks into $\boldsymbol{Z}_{\mathrm{per}}$ from $\boldsymbol{Z}_{\mathrm{sem}}$, as shown in the $I(\boldsymbol{Z}_\bullet, \boldsymbol{Y})$ plots. Thus, our insight from the flattening experiments also translates to image models: $I(\boldsymbol{X}; \boldsymbol{Y})$ is sourced from a different stage of the diffusion process than the one where the model strains to pin down the image manifold. This also explains why CFG is effective in image diffusion models; the CFG vector is strongest in the interval where the scores are well-behaved.

Finally, it is worth considering how this picture extends to higher-resolution image datasets. As image resolution increases, the pixel count grows quadratically, whereas the intrinsic dimensionality of the data manifold grows much more slowly. There are far more nearest-neighbor pixel correlations to resolve, so TC($\boldsymbol{X}$) explodes, while $I(\boldsymbol{X}; \boldsymbol{Y}) \leq S(\boldsymbol{Y})$ grows logarithmically in the number of classes. Thus, the ratio $I(\boldsymbol{X}; \boldsymbol{Y})/S_{\mathrm{NN}}^{\boldsymbol{X}}$ is *smaller*, which strengthens the separability argument. This is already apparent in Fig. 16, where $S_{\mathrm{NN}}^{\boldsymbol{X}}$ for Tiny ImageNet is an order of magnitude larger than that of MNIST, but $I(\boldsymbol{X}; \boldsymbol{Y})$ merely doubles. More details about these experiments are given in Sec. D.

## 6. Conclusion

We have shown that diffusion models learn correlations between the components of a signal $\boldsymbol{X}$ itself, as well as between signals $\boldsymbol{X}$ and $\boldsymbol{Y}$. To reproduce the former, the

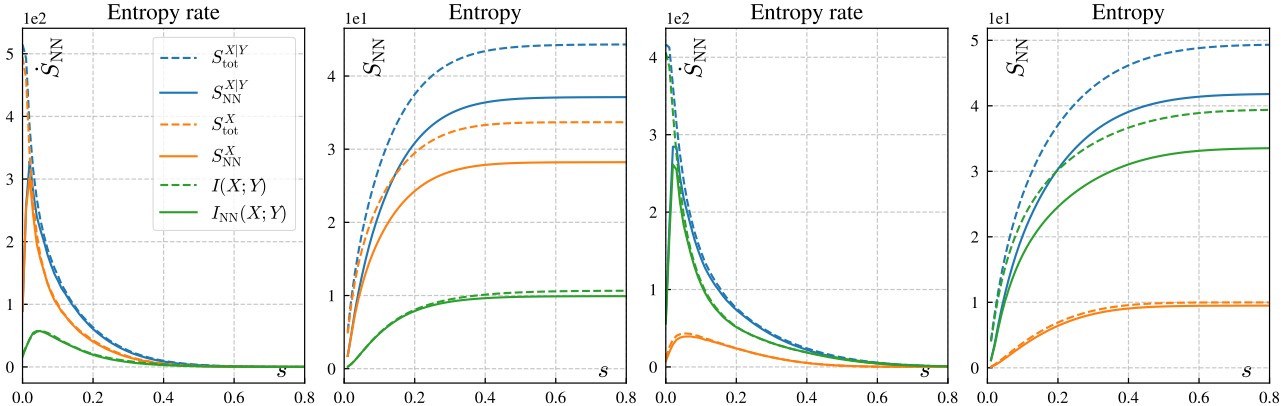

**Figure 4.** Two ways of having a large $S_{\mathrm{NN}}^{\boldsymbol{X}|\boldsymbol{Y}}$. (Left) *Flattening:* $I(\boldsymbol{X};\boldsymbol{Y})$ stays finite while $S_{\mathrm{NN}}^{\boldsymbol{X}}$ blows up due to lower intrinsic dimensionality of $\boldsymbol{X}$. This is a toy model of what happens in image diffusion models. (Right) *Determinism:* $I(\boldsymbol{X};\boldsymbol{Y})$ diverges when $\boldsymbol{Y}$ is strongly correlated with $\boldsymbol{X}$, but $S_{\mathrm{NN}}^{\boldsymbol{X}}$ is under control since its covariance is full-rank. The entropy rate curves are the time derivatives of the corresponding entropy or mutual information. The dashed lines are computed with analytic expressions of the score function, while the solid lines are estimates from the diffusion model (see Sec. D.1). Similar plots for different degrees of flattening and correlation are given in Figs. 11 and 12.

model must absorb $\mathrm{TC}(\boldsymbol{X})$ worth of information, which is the non-trivial part of $S_{\mathrm{tot}}^{\boldsymbol{x}}$. On top of this, an additional $I(\boldsymbol{X};\boldsymbol{Y})$ information must be applied to correlate $\boldsymbol{X}$ with $\boldsymbol{Y}$. When $\boldsymbol{X}$ lives on a low-dimensional manifold, $\mathrm{TC}(\boldsymbol{X})$ is high, but $I(\boldsymbol{X};\boldsymbol{Y})$ is mostly decoupled from this divergence. This explains why CFG works, despite the low ratio of $I(\boldsymbol{X};\boldsymbol{Y})$ to $S_{\mathrm{tot}}^{\boldsymbol{x}}$ for image datasets.

Our findings can provide insight into some practical aspects of diffusion models. First, in latent-space models, the semantic information is compressed into a latent representation, the distribution of which is modeled with diffusion. The perceptual details are outsourced to a separate decoder (Rombach et al., 2022; Vahdat et al., 2021). The efficacy of such models becomes clear in light of our work: by making the diffusion model responsible solely for the semantic reconstruction, the model is spared the informational load of focusing the probability mass onto $\mathcal{M}_{\boldsymbol{X}}$. This makes the overall pipeline much better adapted to generating higher-dimensional data.

Second, distilled diffusion models often struggle to preserve fine-grained details in the images, especially in methods that compress sampling to very few steps (Dong et al., 2025; Dieleman, 2024b). Some approaches mitigate this issue by including an adversarial loss that 'forces the model to directly generate samples that lie on the image manifold, avoiding blurriness' (Sauer et al., 2024). Our arguments suggest that distillation veers off $\mathcal{M}_{\boldsymbol{X}}$ because the distilled model does not capture with sufficient fidelity the information concentrated in $s < \tau$ in Fig. 3. These information-intensive steps coincide with the high-curvature phase of the generative trajectories as they approach the data manifold, which can only be learned correctly with finer dis-

cretization steps (Lee et al., 2023; Chen et al., 2024).

The core message of our paper has a close parallel in the renormalization group (RG) theory from physics. Formally, $D_{\mathrm{KL}}(p_{\mathrm{d}}\|p_{\mathrm{eq}})$ is infinite for images because nearest-neighbor pixels are strongly correlated with one another. It can be shown that the KL between distributions on the space of quantum fields also exhibits such *UV divergences* on account of contact terms that arise in their calculation (Balasubramanian et al., 2015). The connection between diffusion and RG is sharpened in (Cotler & Rezchikov, 2023a), where it is shown that Polchinski's exact RG is equivalent to an optimal transport gradient flow of a similar KL. Furthermore, the semantic vs. perceptual distinction maps to the idea of relevance vs. irrelevance in RG: the semantic details inform the image's class in the same way that relevant operators determine which universality class the system flows to.

**Related work** Several works have studied mutual information in the context of diffusion models (Franzese et al., 2024; Kong et al., 2023; 2024; Wang et al., 2025). Total correlation has been explored in a VAE setting in (Gao et al., 2019). (Stanczuk et al., 2024) proves that diffusion models encode data manifolds, and introduces a method to extract their intrinsic dimension. However, this work is the first to study the interplay between $\mathrm{TC}(\boldsymbol{X})$ and $I(\boldsymbol{X};\boldsymbol{Y})$ in diffusion models, and connect them with the dimensionality of the data manifold. Furthermore, we attach a 'physical' meaning to these quantities, interpreting them as information *stored in the neural network*. This is an extension of the idea introduced in (Premkumar, 2025), where the control effort to transform $p_{\mathrm{eq}} \to p_{\mathrm{d}}$ is related to the information required to locate $p_{\mathrm{d}}$ in an ensemble centered

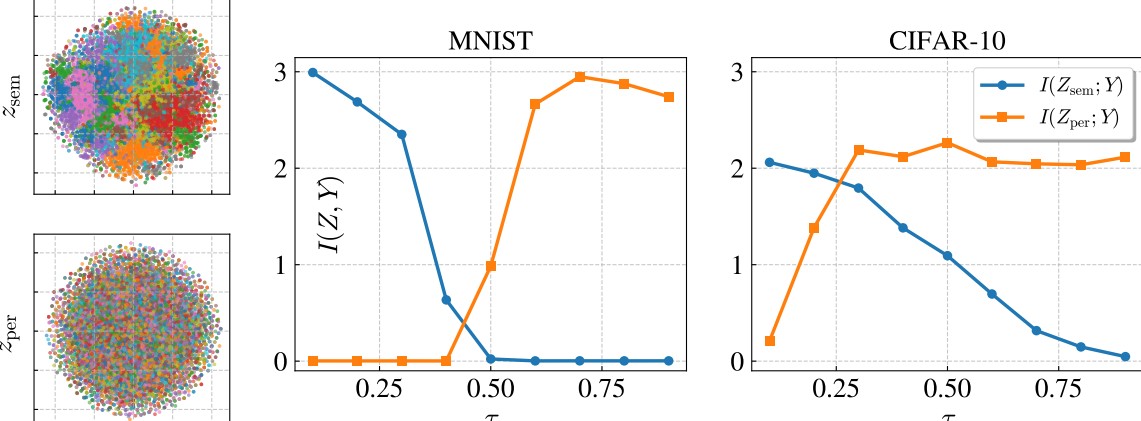

*Figure 5.* Probing the information stored in a diffusion model using a diffusion autoencoder. (Left column) t-SNE plots of $z_{\text{sem}}$ and $z_{\text{per}}$ for MNIST with $\tau = 0.1T$. The former shows discernible clusters corresponding to the different digits, while the latter has no such structure. This indicates that the information collected from $s \in (0, \tau)$ contains little to no information about the digits. (Right) The correlation between the learned latents and the true labels, as quantified by $I(Z_\bullet; Y)$. See Fig. 17.

at $p_{\text{eq}}$. (Stančević et al., 2025; Stančević & Ambrogioni, 2026) also study diffusion dynamics through the lens of entropy and information flow. In (Handke et al., 2026), which appeared concurrently with this work, class-conditional entropy is used to study the emergence of semantic structure in diffusion models and how guidance affects this process. (Xiang et al., 2023; Chen et al., 2025) argue that diffusion models internally separate semantic from perceptual content along the noise scale, by probing intermediate U-net activations. Our DAE experiments are complementary: we treat the diffusion decoder as a black box and study how it imprints semantic and perceptual information onto a latent code learned by an external encoder. Some connections between RG and diffusion models are explored in (Cotler & Rezchikov, 2023b).

ACKNOWLEDGMENTS

We thank Luca Ambrogioni, Aram-Alexandre Pooladian, Daniel Green, Kshitij Gupta, Tongyan Lin, and Sandip Roy for helpful discussions. This paper was originally inspired by a connection between probabilistic modeling and the Kelly criterion (Kelly, 1956), which arose after a conversation with William Cottrell. Daniele Rogantini, Georgios Valogiannis, and Chun-Hao To provided the initial motivation to compile these ideas into a paper. Experiments in this work were carried out in the Bouchet cluster at the Yale Center for Research Computing (YCRC). The author was supported through an AI seed grant from Yale at this time.

## Impact Statement

This paper presents work whose goal is to advance the field of machine learning. There are many potential societal con-

sequences of our work, none of which we feel must be specifically highlighted here.

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

# A. Information Theory

## A.1. Mutual Information: A Primer

The mutual information between two random variables quantifies the reduction in uncertainty of one variable given the value of the other variable. If $X$ and $Y$ are the random variables in question, then the mutual information $I(X;Y)$ is the information gained about $X$ through a measurement of $Y$. If $p(x,y)$ is the joint distribution of $X$ and $Y$ and $p(x)$ and $p(y)$ are the marginals,

$$
\begin{aligned}
I(X;Y) &:= D_{\mathrm{KL}}\left(p(x,y)\|p(x)p(y)\right) \\
&= S(X) - S(X|Y) = S(Y) - S(Y|X).
\end{aligned}
\tag{18}
$$

By construction, $I(X;Y)$ is symmetric in its arguments, and it is non-negative. Mutual information captures *all* forms of statistical dependence, not just linear ones. That said, it is easier to develop some intuition for $I(X;Y)$ by considering a simple linear model

$$
Y = aX + \varepsilon,
\tag{19}
$$

where $X \sim \mathcal{N}(0, \sigma_X^2), \varepsilon \sim \mathcal{N}(0, \sigma_\varepsilon^2)$, and $a$ is a real constant (Reeves et al., 2018). It is easy to see that

$$
Y|X = x \sim \mathcal{N}(ax, \sigma_\varepsilon^2),
\tag{20a}
$$

$$
Y \sim \mathcal{N}(0, a^2\sigma_X^2 + \sigma_\varepsilon^2),
\tag{20b}
$$

$$
X|Y = y \sim \mathcal{N}\left(\frac{a\sigma_X^2}{a^2\sigma_X^2 + \sigma_\varepsilon^2}y; \frac{\sigma_X^2\sigma_\varepsilon^2}{a^2\sigma_X^2 + \sigma_\varepsilon^2}\right),
\tag{20c}
$$

where Eq. (20a) follows from the fact that $Y$ is a scaled version of $X$ with some noise added to it, and Eq. (20b) is obtained by marginalizing this distribution over $X$. With these distributions, Eq. (20c) can be derived using Bayes' rule. Notice that $X|Y$ has a smaller variance than $X$. This is what we mean when we say $p(x|y)$ is 'narrower' than $p(x)$ (see Fig. 6), although for general distributions this is only true *on average*—there can be cases where the conditional is broader than the marginal for some $y$.

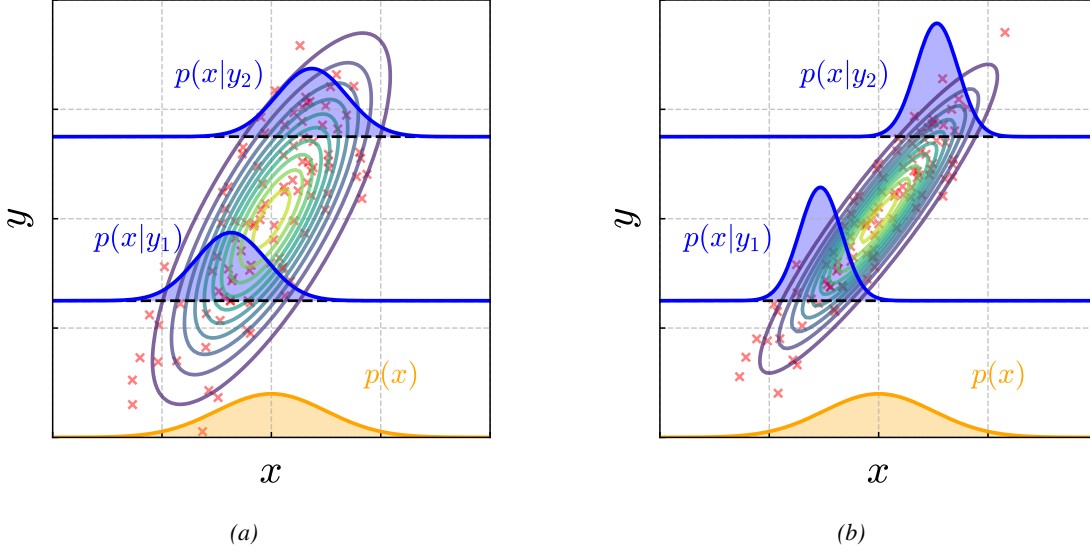

*(a)*           *(b)*

*Figure 6.* The linear Gaussian model from Eq. (19) with (a) higher noise/larger $\sigma_\varepsilon$, and (b) lower noise/smaller $\sigma_\varepsilon$. The blue curves are the conditionals $X|Y = y$ for some $y$, and the orange curve is the marginal over $X$. Notice how the conditionals have a tighter variance compared to the marginal. The contours are surfaces over constant probability in the joint distribution, and the red markers are some samples.

The main point is that knowing $Y$ dispels some of the uncertainty in $X$. That is, $X|Y = y$ has a lower entropy on average than $X$,

$$
S(X|Y) = \int \mathrm{d}y\, p(y)S(X|Y = y) = -\int \mathrm{d}y\, p(y)\int \mathrm{d}x\, p(x|y)\log p(x|y) \leq S(X).
\tag{21}
$$

Mutual information is the difference between these two entropies. We can compute the latter explicitly from Eqs. (20a) and (20b),

$$I(X;Y) = S(Y) - S(Y|X) = \frac{1}{2} \log \frac{\text{Var}(Y)}{\text{Var}(Y|X)} = \frac{1}{2} \log \left( 1 + \frac{a^2 \sigma_X^2}{\sigma_\varepsilon^2} \right). \tag{22}$$

Notice that $I(X;Y) \to 0$ as $\sigma_\varepsilon \gg \sigma_X$, since the $X$ signal is drowned out by the noise in this regime. In the opposite limit, when noise is very weak, $X$ and $Y$ are very strongly correlated and $I(X;Y)$ grows. If $X$ and $Y$ were discrete random variables, $I(X;Y)$ would have saturated at $S(X)$. However, in the continuous case, mutual information can diverge to infinity, which is the same pathology shared by differential entropy (Cover & Thomas, 2006, Chap. 8). Indeed, in the noiseless limit $p(x,y)$ collapses onto the line $y = ax$ whereas the product $p(x)p(y)$ spreads mass over the whole plane, so $p(x,y)$ is singular with respect to $p(x)p(y)$ in Eq. (18) (see Fig. 12).

This peculiar behavior of $I(X;Y)$ in the continuous case is reminiscent of the singular growth in entropy in image diffusion models as $t \to T$ (see Fig. 16). It is in fact the same phenomenon, which arises when the joint distribution converges on a lower dimensional manifold in the noiseless limit (see Fig. 7). In diffusion models the piece that becomes singular is the total correlation term in Eq. (7), TC($\boldsymbol{X}$), which is a generalization of mutual information. We shall show in Sec. E.3 that the re-establishment of perceptual detail in the images coincides with a sharp peak of the neural entropy rate at the final stages of the generative process. Correlations between nearby pixels must be made very tight to get these small-scale details correct, which forces the image vector to track a manifold of lower dimensions, resulting in a divergent TC($\boldsymbol{X}$). This is why the small details of the image take up a sizable portion of the total information budget.

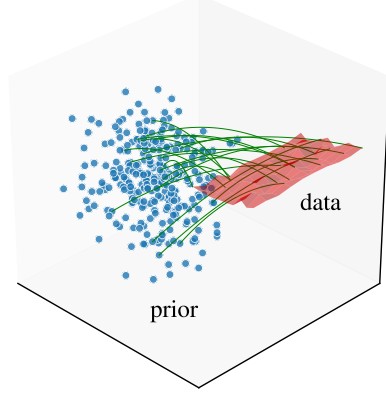

*Figure 7*

The divergence in $I(X;Y)$ and TC($\boldsymbol{X}$) cannot be eliminated by a change of basis. It is easier to understand this for $I(X;Y)$ first, and then extend the intuition to TC($\boldsymbol{X}$). Consider the model in Eq. (19), rotated to a new basis $(u,v)$ where the $u$-axis is the line $y = ax$ and the $v$-axis is orthogonal to it. As $\sigma_\varepsilon \to 0$, the joint distribution $p(u,v)$ collapses onto the $u$-axis. However, in the product of marginals $p(u)p(v)$, the density $p(v)$ is supported entirely on a set of measure zero. Thus, $I(U;V)$ is still infinite. In the same way, TC($\boldsymbol{X}$) remains divergent if $\boldsymbol{X}$ lives on a lower dimensional manifold, even if there is a local set of coordinates that perfectly aligns with the surface of the manifold.

### A.2. The Information Bottleneck

Consider the problem of building a classifier that maps an input $X$ to a label $Y$, where $Y$ is now a discrete random variable. We would like to find a representation $Z$ of $X$ that captures all information in $X$ that is relevant to predicting $Y$, while discarding the superfluous details. This is the viewpoint formalized by the information bottleneck, where the optimal assignment from $X$ to $Z$ is obtained by varying the stochastic map $q(z|x)$ to solve

$$\min_{q(z|x)} I(X;Z) - \gamma^{-1} I(Z;Y), \tag{23}$$

with $\gamma > 0$. A rigorous derivation of this functional is given in (Tishby et al., 2000), but the intuition behind it is simple: minimizing $I(X;Z)$ maps a wide range of $X$ values to a narrow range of the latent variable $Z$, and maximizing $I(Z;Y)$ creates a map between $Z$ and $Y$ where knowing $Z$ almost completely determines $Y$. This forces $q(z|x)$ to only encode features from $X$ into $Z$ that most strongly associate with the class label $Y$, which is a form of selective compression. By adjusting $\gamma$ we can set the tradeoff between compression and information preservation—setting $\gamma = 0$ collapses $q(z|x)$ to a single point, whereas $\gamma \to \infty$ pushes maximal detail from $X$ to $Z$.

In general, $Y$ can be any random variable, including a reconstruction of $X$ itself. This is an autoencoder. However, there are a few subtle differences between Eq. (23) and the standard formulation of autoencoders (Kingma & Welling, 2014; Alemi et al., 2016). To see the connection, we expand Eq. (23) to

$$\min_{q(z|x)} S(Z) - S(Z|X) - \gamma^{-1}(S(Y) - S(Y|Z)). \tag{24}$$

The entropy of $Y$ is independent of the encoder, so we can drop it from the objective (cf. Eq. (70)). We can introduce a regularization term to make the $Z$ distribution close to a prior $p(z)$, like a standard normal distribution. This controls the entropy of $Z$ from becoming too large as we vary $q(z|x)$. With these modifications Eq. (24) becomes

$$\min_{q(z|x)} \mathbb{E}_{x,y} \mathbb{E}_{q(z|x)}[- \log p(y|z)] + \gamma \mathbb{E}_x D_{\mathrm{KL}}(q(z|x) \,\|\, p(z)). \tag{25}$$

Setting $Y = X$, we recover the negative ELBO from Eq. (67).

## B. Stochastic Control

In Sec. 2, we started with a formula for $S_{\mathrm{tot}}$ Eq. (3), which is derived in (Premkumar, 2025) using stochastic optimal control theory. In this section, we derive several known results about diffusion models following the same approach. This allows us to see how the Miyasawa/Tweedie relations, the denoising score matching objective, and the mutual information formula of (Franzese et al., 2024), all spring from the same underlying optimization principle.

### B.1. Log Likelihood Bound

In Sec. 2 we introduced the forward and reverse SDEs, Eqs. (1) and (2), which take $p_{\mathrm{d}} \to p_0$ and back. If we replace the reverse process with an SDE (see Fig. 19)

$$\mathrm{d}\boldsymbol{X}_t = -\mathfrak{u}(\boldsymbol{X}_t, t)\mathrm{d}t + \sigma(T - t)\mathrm{d}\boldsymbol{B}_t, \tag{26}$$

and distribution $p_{\mathfrak{u}}(\cdot, 0)$ at time $t = 0$ evolves to $p_{\mathfrak{u}}(\cdot, T)$ at $t = T$, the log density of which is bound as (see Premkumar, 2025, App. E.2)

$$-\log p_{\mathfrak{u}}(\boldsymbol{x}, T) \le \mathbb{E}\left[\left(\int_0^T \mathrm{d}s \frac{\|b_+ - \mathfrak{u}\|^2}{2\sigma^2} + \mathfrak{u} \cdot \nabla \log p(\tilde{\boldsymbol{x}}_s, s|\tilde{\boldsymbol{x}}_0, 0)\right) - \log p_{\mathfrak{u}}(\tilde{\boldsymbol{x}}_T, 0)\middle| \tilde{\boldsymbol{X}}_0 = \boldsymbol{x}\right]. \tag{27}$$

The expectation value is taken over all trajectories generated by Eq. (1), starting at $\tilde{\boldsymbol{X}}_0 = \boldsymbol{x}$. This inequality, which can be derived from optimal control theory (Pavon, 1989), or the Feynman-Kac formula (Ge & Jiang, 2008; Huang et al., 2021), will be the starting point of many of our derivations. Completing the square, the bound can be written as

$$-\log p_{\mathfrak{u}}(\boldsymbol{x}, T) \le \int_0^T \mathrm{d}s \frac{1}{2\sigma^2} \mathbb{E}\left[\left\|b_+ - \sigma^2 \nabla \log p(\tilde{\boldsymbol{x}}_s, s|\tilde{\boldsymbol{x}}_0, 0) - \mathfrak{u}\right\|^2 \middle| \tilde{\boldsymbol{X}}_0 = \boldsymbol{x}\right] \tag{28}$$
$$- \int_0^T \mathrm{d}s\, \mathbb{E}\left[\frac{\sigma^2}{2}\left\|\nabla \log p(\tilde{\boldsymbol{x}}_s, s|\tilde{\boldsymbol{x}}_0, 0)\right\|^2 + \nabla \cdot b_+ \middle| \tilde{\boldsymbol{X}}_0 = \boldsymbol{x}\right] + S_0.$$

We have replaced $\mathbb{E}[-\log p_{\mathfrak{u}}(\tilde{\boldsymbol{x}}_T)]$ with the negative differential entropy $S_0 := \mathbb{E}_{p_0}[-\log p_0]$ by choosing $p_{\mathfrak{u}}(\cdot, 0)$ to be $p_0(\cdot)$ and noting that, to a very good approximation, $\tilde{\boldsymbol{x}}_T$ would be distributed as $p_0$ irrespective of the $\boldsymbol{x}$ at which it started.

Averaging Eq. (28) over the data distribution $p_{\mathrm{d}}$ yields an upper bound on the cross-entropy between $p_{\mathrm{d}}$ and the reconstructed distribution $p_{\mathfrak{u}}(\cdot, T)$. In a diffusion model, a neural network parametrizes the *control* $\mathfrak{u}$, which affects only one term in the bound. Therefore, minimizing the cross-entropy is equivalent to minimizing the *denoising* objective

$$\mathbb{E}_{\boldsymbol{X}}[-\log p_{\mathfrak{u}}(\boldsymbol{x}, T)] + c \le \int_0^T \mathrm{d}s \frac{1}{2\sigma^2} \mathbb{E}_{\boldsymbol{X}, \tilde{\boldsymbol{X}}_s}\left[\left\|b_+ - \sigma^2 \nabla \log p(\tilde{\boldsymbol{x}}_s, s|\boldsymbol{x}, 0) - \mathfrak{u}\right\|^2\right] := \mathcal{L}_{\mathrm{D}}, \tag{29}$$

where $c$ denotes the $\mathfrak{u}$-independent terms from Eq. (28), averaged over $\boldsymbol{X}$. In an entropy-matching model $\mathfrak{u} = -b_+ - \sigma^2 \boldsymbol{e_\theta}$, so the denoising entropy-matching objective is (cf. Eq. (68))

$$\mathbb{E}_{\boldsymbol{X}}[-\log p_{\boldsymbol{\theta}}(\boldsymbol{x}, T)] + c \le$$
$$\int_0^T \mathrm{d}s \frac{\sigma^2}{2} \mathbb{E}_{\boldsymbol{X}, \tilde{\boldsymbol{X}}_s}\left[\left\|\nabla \log p_{\mathrm{eq}}(\tilde{\boldsymbol{x}}_s) - \nabla \log p(\tilde{\boldsymbol{x}}_s, s|\boldsymbol{x}, 0) + \boldsymbol{e_\theta}(\tilde{\boldsymbol{x}}_s, s)\right\|^2\right] := \mathcal{L}_{\mathrm{DEM}}. \tag{30}$$

A similar objective can be derived for score-matching models by setting $\mathfrak{u} = b_+ - \sigma^2 \boldsymbol{s_\theta}$ in Eq. (29), or equivalently, $\boldsymbol{s_\theta} = \nabla \log p_{\mathrm{eq}} + \boldsymbol{e_\theta}$ in Eq. (28) (Song et al., 2021a; Kingma & Welling, 2014).

## B.2. Optimal Control and Regression

The bound in Eq. (27) is saturated by the *optimal control*,

$$\mathfrak{u}_\star = b_+ - \sigma^2 \nabla \log p, \tag{31}$$

which turns Eq. (26) into the reverse SDE Eq. (2), and the cross-entropy in Eq. (29) reaches its minimum value of $\mathbb{E}_{\boldsymbol{X}}[-\log p_{\mathrm{d}}]$ (Pavon, 1989; Huang et al., 2021). But that also means $\mathfrak{u}_\star$ minimizes the denoising objective $\mathcal{L}_{\mathrm{D}}$,

$$\mathfrak{u}_\star = \arg\min_{\mathfrak{u}(\cdot,\cdot)} \mathcal{L}_{\mathrm{D}}. \tag{32}$$

We apply Theorem 14.60 of (Rockafellar & Wets, 1998), which guarantees that the minimization of a time-integrated convex loss functional is achieved by pointwise minimization of the integrand. Briefly, given a normal integrand $\mathcal{J}$ and a measurable weight function $\lambda(s) \geq 0$, the minimization of $\mathcal{J}$ over the space $\chi$ of measurable functions $f : [0, T] \to \mathbb{R}^{D_{\boldsymbol{x}}}$ is

$$f_\star \in \arg\min_{f(\cdot)\in\chi} \int_0^T \mathrm{d}s\, \lambda(s) \mathcal{J}(s, f(s)) = f_\star(s) \Leftrightarrow \arg\min_{f \in \mathbb{R}^{D_{\boldsymbol{x}}}} \mathcal{J}(s, f), \text{ for almost every } s \in [0, T]. \tag{33}$$

This allows us to analyze the denoising objective independently at each time $s$, which reduces Eq. (32) to a family of decoupled conditional mean regression problems, each minimizing the expected squared deviation at time $s$ (see Sec. 1.5.5 of Bishop, 2006):

$$\mathfrak{u}_\star = \arg\min_{\mathfrak{u}\in\mathbb{R}^{D_{\boldsymbol{x}}}} \int \mathrm{d}\tilde{\boldsymbol{x}}_s \int \mathrm{d}\boldsymbol{x}\, p(\tilde{\boldsymbol{x}}_s, \boldsymbol{x}) \|b_+(\tilde{\boldsymbol{x}}_s) - \sigma^2 \nabla \log p(\tilde{\boldsymbol{x}}_s, s|\boldsymbol{x}, 0) - \mathfrak{u}(\tilde{\boldsymbol{x}}_s, s)\|^2$$

$$= b_+(\tilde{\boldsymbol{x}}_s) - \sigma^2 \mathbb{E}_{\boldsymbol{x}\sim p(\boldsymbol{x}|\tilde{\boldsymbol{x}}_s)} \nabla \log p(\tilde{\boldsymbol{x}}_s, s|\boldsymbol{x}, 0). \tag{34}$$

Comparing with Eq. (31), we conclude that

$$\nabla_{\tilde{\boldsymbol{x}}_s} \log p(\tilde{\boldsymbol{x}}_s, s) = \mathbb{E}_{\boldsymbol{x}\sim p(\boldsymbol{x}|\tilde{\boldsymbol{x}}_s)} \nabla_{\tilde{\boldsymbol{x}}_s} \log p(\tilde{\boldsymbol{x}}_s, s|\boldsymbol{x}, 0). \tag{35}$$

If the perturbation kernel is Gaussian,[2]

$$\nabla_{\tilde{\boldsymbol{x}}_s} \log p(\tilde{\boldsymbol{x}}_s, s|\boldsymbol{x}, 0) = -\frac{\tilde{\boldsymbol{x}}_s - \mu(s)\boldsymbol{x}}{\Sigma(s)} \tag{36}$$

$$\overset{35}{\Longrightarrow} \nabla_{\tilde{\boldsymbol{x}}_s} \log p(\tilde{\boldsymbol{x}}_s, s) = -\frac{\tilde{\boldsymbol{x}}_s - \mu(s)\mathbb{E}[\boldsymbol{x}|\tilde{\boldsymbol{x}}_s]}{\Sigma(s)}, \tag{37}$$

which is the Miyasawa relation/Tweedie's formula (Miyasawa, 1961; Efron, 2011). In simple terms, this relation states that at any given $s$, the ideal score is a vector pointing from $\tilde{\boldsymbol{x}}_s$ toward the denoised mean $\mathbb{E}[\boldsymbol{x}|\tilde{\boldsymbol{x}}_s]$, scaled by a forward factor. As mentioned above, Eq. (29) turns into an equality under Eq. (31), with $p_{\mathfrak{u}_\star}(\cdot, T) = p_{\mathrm{d}}(\cdot)$,

$$\mathbb{E}_{\boldsymbol{X}}[-\log p_{\mathrm{d}}(\boldsymbol{x})] + c = \int_0^T \mathrm{d}s\, \frac{\sigma^2}{2} \mathbb{E}_{\boldsymbol{X},\tilde{\boldsymbol{X}}_s} \left[\|\nabla \log p(\tilde{\boldsymbol{x}}_s, s) - \nabla \log p(\tilde{\boldsymbol{x}}_s, s|\boldsymbol{x}, 0)\|^2\right]. \tag{38}$$

For the Gaussian kernel, we can substitute Eqs. (36) and (37) and write this as (Kingma et al., 2021)

$$\mathbb{E}_{\boldsymbol{X}}[-\log p_{\mathrm{d}}(\boldsymbol{x})] + c = \int_0^T \mathrm{d}s\, \frac{\sigma(s)^2}{2} \frac{\mu(s)^2}{\Sigma(s)^2} \mathbb{E}_{\boldsymbol{X},\tilde{\boldsymbol{X}}_s} \left[\|\mathbb{E}[\boldsymbol{x}|\tilde{\boldsymbol{x}}_s] - \boldsymbol{x}\|^2\right]$$

$$=: \int_0^T \mathrm{d}s\, B(s) \mathbb{E}_{\boldsymbol{X},\tilde{\boldsymbol{X}}_s} \left[\|\hat{\boldsymbol{x}}(\tilde{\boldsymbol{x}}_s) - \boldsymbol{x}\|^2\right]. \tag{39}$$

In the last step, we have collected the time-dependent prefactor into a single function $B(s)$, and defined $\hat{\boldsymbol{x}}(\tilde{\boldsymbol{x}}_s) := \mathbb{E}[\boldsymbol{x}|\tilde{\boldsymbol{x}}_s]$.

---

[2] The kernel is Gaussian for Ornstein-Uhlenbeck processes, which have the form (Karras et al., 2022)

$$\mathrm{d}\tilde{\boldsymbol{X}}_s = \phi(s)\tilde{\boldsymbol{X}}_s \mathrm{d}s + \sigma(s)\mathrm{d}\boldsymbol{B}_s.$$

The perturbation kernel of this SDE is

$$p(\tilde{\boldsymbol{x}}_s, s|\boldsymbol{x}, 0) = \mathcal{N}\left(\tilde{\boldsymbol{x}}_s; \mu(s)\boldsymbol{x}, \Sigma(s)I\right),$$

where

$$\mu(s) = \exp\left(\int_0^s \mathrm{d}\bar{s}\phi(\bar{s})\right), \qquad \Sigma(s) = \mu(s)^2 \int_0^s \mathrm{d}\bar{s}\frac{\sigma(\bar{s})^2}{\mu(\bar{s})^2}.$$

### B.3. Reweighted Objective

Notice that the choice of weight function $\lambda(s)$ in Eq. (33) does not affect the pointwise minimization that leads to Eq. (35). This provides a theoretical justification of 'variance dropping' in practical denoising objectives such as Eq. (30) (Ho et al., 2020). That is, $\mathcal{L}_{\text{DEM}}$ is replaced by the Monte Carlo average

$$T\,\mathbb{E}_{\boldsymbol{x}\sim p_{\mathrm{d}}}\mathbb{E}_{s\sim\mathcal{U}(0,T)}\left[\lambda(s)\,\mathbb{E}_{\tilde{\boldsymbol{x}}_s\sim p(\tilde{\boldsymbol{x}}_s|\boldsymbol{x})}\left\|\nabla\log p_{\text{eq}}^{(s)}(\tilde{\boldsymbol{x}}_s)+\boldsymbol{e}_{\boldsymbol{\theta}}(\tilde{\boldsymbol{x}}_s,s)-\nabla\log p(\tilde{\boldsymbol{x}}_s,s|\boldsymbol{x},0)\right\|^2\right], \tag{40}$$

where $\mathcal{U}$ is the uniform distribution over $(0,T)$ and $\lambda(s)$ is not necessarily $\sigma^2(s)/2$. Dropping the variance means setting $\lambda(s)=1$. In principle $\nabla\log p_{\text{eq}}+\boldsymbol{e}_{\boldsymbol{\theta}}$ still recovers the optimal score from Eq. (35), but empirical observations show that alternative weighting schemes improve numerical stability and reduce gradient variance (Song et al., 2021a; Kingma & Gao, 2023). We used $\lambda(s)=1$ in all our image diffusion models, including the DAE decoders. However, when applying Eq. (12) to estimate the mutual information we noticed that choosing $\lambda(s)=\sigma(s)^2/2$ gives slightly more accurate results.

## C. Mutual Information from Diffusion

### C.1. MINDE

In Sec. 3 we discussed Eq. (12), a formula for mutual information originally derived in (Franzese et al., 2024). We will give a derivation of this result using Eq. (27). Setting the control to its optimal value, Eq. (31), and integrating by parts,

$$-\log p_{\mathrm{d}}(\boldsymbol{x})=\mathbb{E}\left[\left(\int_0^T \mathrm{d}s\,\frac{\sigma^2}{2}\left\|\nabla\log p\right\|^2-\nabla\cdot(b_+-\sigma^2\nabla\log p)\right)-\log p_0(\tilde{\boldsymbol{x}}_T)\bigg|\tilde{\boldsymbol{X}}_0=\boldsymbol{x}\right]. \tag{41}$$

Here $\nabla\log p\equiv\nabla\log p(\tilde{\boldsymbol{x}}_s,s)$. Averaging this over $\boldsymbol{X}$ yields the entropy of $p_{\mathrm{d}}(\boldsymbol{x})$ (cf. Eq. (28)),

$$S(\boldsymbol{X})=\mathbb{E}_{\boldsymbol{X}}[-\log p_{\mathrm{d}}(\boldsymbol{x})]=\int_0^T \mathrm{d}s\,\mathbb{E}_{\tilde{\boldsymbol{X}}_s,\boldsymbol{X}}\left[\frac{\sigma^2}{2}\left\|\nabla\log p\right\|^2-\nabla\cdot(b_+-\sigma^2\nabla\log p)\right]+S_0. \tag{42}$$

A similar formula can be derived for $S(\boldsymbol{X}|\boldsymbol{Y})$, by changing $\nabla\log p(\tilde{\boldsymbol{x}}_s,s)\to\nabla\log p(\tilde{\boldsymbol{x}}_s,s|\boldsymbol{y})$ and averaging over $\boldsymbol{y}$ also. Mutual information is just the difference between the two,

$$
\begin{aligned}
I(\boldsymbol{X};\boldsymbol{Y}) &= S(\boldsymbol{X})-S(\boldsymbol{X}|\boldsymbol{Y})\\
&= \mathbb{E}_{\boldsymbol{Y}}\left[\int_0^T \mathrm{d}s\,\frac{\sigma^2}{2}\mathbb{E}_{\tilde{\boldsymbol{X}}_s,\boldsymbol{X}|\boldsymbol{y}}\left[\left\|\nabla\log p(\tilde{\boldsymbol{x}}_s,s)\right\|^2-\left\|\nabla\log p(\tilde{\boldsymbol{x}}_s,s|\boldsymbol{y})\right\|^2\right.\right.\\
&\qquad\qquad\qquad\left.\left.+2\nabla\cdot(\nabla\log p(\tilde{\boldsymbol{x}}_s,s)-\nabla\log p(\tilde{\boldsymbol{x}}_s,s|\boldsymbol{y}))\right]\right]\\
&\stackrel{\text{IBP}}{=} \mathbb{E}_{\boldsymbol{Y}}\left[\int_0^T \mathrm{d}s\,\frac{\sigma^2}{2}\mathbb{E}_{\tilde{\boldsymbol{X}}_s,\boldsymbol{X}|\boldsymbol{y}}\left[\left\|\nabla\log p(\tilde{\boldsymbol{x}}_s,s)\right\|^2-\left\|\nabla\log p(\tilde{\boldsymbol{x}}_s,s|\boldsymbol{y})\right\|^2\right.\right.\\
&\qquad\qquad\qquad\left.\left.-2(\nabla\log p(\tilde{\boldsymbol{x}}_s,s)-\nabla\log p(\tilde{\boldsymbol{x}}_s,s|\boldsymbol{y}))\cdot\nabla\log p(\tilde{\boldsymbol{x}}_s,s|\boldsymbol{y})\right]\right]\\
&= \mathbb{E}_{\boldsymbol{Y}}\left[\int_0^T \mathrm{d}s\,\frac{\sigma^2}{2}\mathbb{E}_{\tilde{\boldsymbol{X}}_s,\boldsymbol{X}|\boldsymbol{y}}\left[\left\|\nabla\log p(\tilde{\boldsymbol{x}}_s,s|\boldsymbol{y})-\nabla\log p(\tilde{\boldsymbol{x}}_s,s)\right\|^2\right]\right].
\end{aligned}
$$

$$ \tag{43} $$
$$ \tag{44} $$

This is Eq. (12). We have assumed that $T$ is sufficiently large that $S_0$ is nearly the same in both cases. We also partitioned the expectation over $\boldsymbol{X}$ into an average over $\boldsymbol{X}|\boldsymbol{Y}=\boldsymbol{y}$ (shortened to $\boldsymbol{X}|\boldsymbol{y}$) and over $\boldsymbol{Y}$ separately. This is why integration by parts in Eq. (43) produced the conditional score term.

It should be noted that Eq. (44) is a generalized version of Eq. (5) that appears in (Kong et al., 2024), or Eq. (2) from (Dewan et al., 2024), which holds only for Gaussian pertubation kernels. This relation is derived from the I-MMSE formula that appears in (Guo et al., 2005; Kong et al., 2023), which relates the mutual information between variables $\boldsymbol{X}$ and $\boldsymbol{Y}=\sqrt{\text{snr}}\boldsymbol{X}+\boldsymbol{\varepsilon}$, where $\boldsymbol{\varepsilon}\sim\mathcal{N}(0,I)$, to the minimium mean-squared error of estimating $\boldsymbol{X}$ from $\boldsymbol{Y}$.

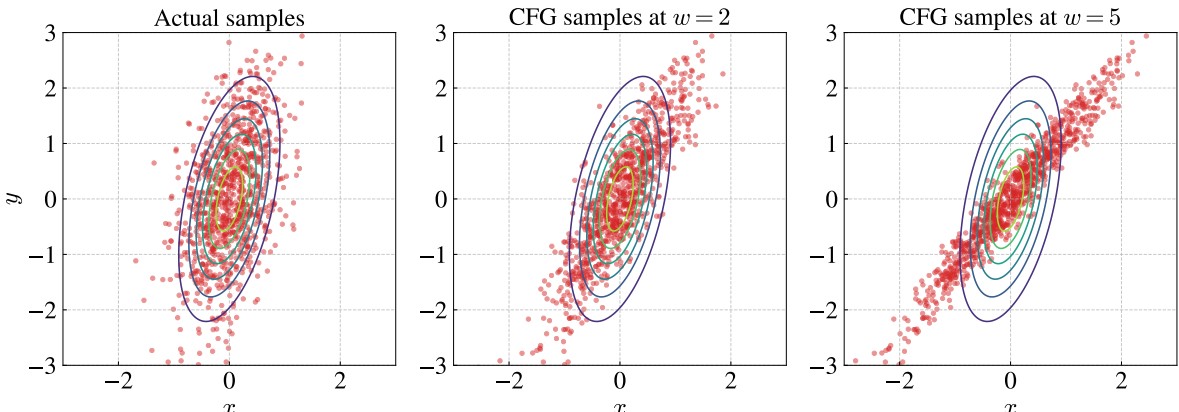

*Figure 8.* Samples generated by a CFG-style modification to the conditional score $\nabla \log p(x_t, t|y)$ of a joint Gaussian (cf. Eqs. (15) and (19)). CFG strengthens the correlation between $X$ and $Y$, increasing their mutual information. But it also alters the relationship between them.

## C.2. Guidance

The expectation value in Eq. (42) is taken over $p(\tilde{\boldsymbol{x}}_s, s)$, which is the density forward evolved from the marginal $p_{\mathrm{d}}(\boldsymbol{x})$. To compute the mutual information between $\boldsymbol{Y}$ and the CFG-generated $\boldsymbol{X}$, we need to compute $S(\boldsymbol{X}|\boldsymbol{Y})_{\mathrm{CFG}}$. The CFG modification from Eq. (15) is equivalent to replacing the score with

$$\nabla \log p(\boldsymbol{x}_t, t|\boldsymbol{y}) + w\boldsymbol{s}_{\mathrm{cl}}(\boldsymbol{y}, t) = \nabla \log \left( \frac{p(\boldsymbol{x}_t, t|\boldsymbol{y})^{1+w}}{p(\boldsymbol{x}_t, t)^w} \right) \tag{45}$$

in Eq. (2). But the resulting SDE is not the reversal of *anything* (Zheng & Lan, 2024; Bradley & Nakkiran, 2024). That is, there is no forward diffusion process for which the intermediate density is $p(\tilde{\boldsymbol{x}}_s, t|\boldsymbol{y})^{1+w}/p(\boldsymbol{x}_t, t)^w$. Samples of this distribution at $t = T$ are different from those of the true $p_{\mathrm{d}}$ (see Fig. 8). Therefore, we cannot write down an expression for $S(\boldsymbol{X}|\boldsymbol{Y})_{\mathrm{CFG}}$ along the lines of Eq. (42), so we find no expression analogous to Eq. (44) for mutual information under CFG.

## D. Experiments

We provide empirical evidence of the following claims made in the main text:

1. Conditional diffusion models store an additional amount of information, equal to the mutual information between $\boldsymbol{X}$ and $\boldsymbol{Y}$ (see Sec. 2 and Fig. 12). In nats,

$$I(\boldsymbol{X}; \boldsymbol{Y}) = S_{\mathrm{tot}}^{\boldsymbol{X}|\boldsymbol{Y}} - S_{\mathrm{tot}}^{\boldsymbol{X}} \approx S_{\mathrm{NN}}^{\boldsymbol{X}|\boldsymbol{Y}} - S_{\mathrm{NN}}^{\boldsymbol{X}}. \tag{46}$$

2. Neural entropy and $I(\boldsymbol{X}; \boldsymbol{Y})$ grow rapidly as $\boldsymbol{X}$ and $\boldsymbol{Y}$ become more strongly correlated, or when $\boldsymbol{X}$ is low-dimensional. See discussions at the end of Secs. A.1 and 3 and Figs. 11 and 12.

3. CFG increases the mutual information between $\boldsymbol{X}$ and $\boldsymbol{Y}$. See Sec. 3 and Fig. 10.

4. For images, the total information content is dominated by perceptual detail, which erodes rapidly in the first few forward diffusion steps. See Sec. E.2 and Figs. 15 and 16.

5. The perceptual information is largely the same for different image classes. Semantic structure is more closely correlated with the labels. See Sec. E.3 and Fig. 17.

The first three points can be demonstrated with a simple Gaussian model, like the one discussed in Sec. A.1. The mutual information and scores are known analytically, which allows us to compare the theoretical values of different entropies

with their estimates from practical diffusion models. Image models are studied in Sec. E, by embedding them inside a DAE.

Our diffusion models used a U-net with self-attention layers (Ho & Salimans, 2022; Salimans & Ho, 2022), and were trained on H200 GPUs with 140 GB of memory. We used JAX/Flax as our ML framework (Bradbury et al., 2018), and trained our image models on the MNIST, CIFAR-10, and Tiny ImageNet datasets (LeCun et al., 1998; Krizhevsky, 2009; Deng et al., 2009). The latter is a 200-class subset of ImageNet with 64×64 resolution images. The Variance Preserving (VP) process was used in all experiments with diffusion models, for which $b_+(\tilde{x}_s, s) = -\beta(s)\tilde{x}_s/2$ and $\sigma(s) = \sqrt{\beta(s)}$ in Eq. (1) (Sohl-Dickstein et al., 2015; Song et al., 2021b). The code is available on GitHub.

## D.1. A Joint Gaussian Model

We revisit the linear model from Eq. (19), generalizing it to higher dimensional random variables $X \in \mathbb{R}^{D_X}$, $Y \in \mathbb{R}^{D_Y}$. That is,

$$Y = AX + \varepsilon, \tag{47}$$

where $X \sim \mathcal{N}(0, \Sigma_X)$ and $\varepsilon \sim \mathcal{N}(0, \Sigma_\varepsilon)$. The joint vector $R := (X, Y)^\top$ is also Gaussian distributed, with zero mean, and covariance

$$\Sigma_R = \begin{pmatrix} \Sigma_X & \Sigma_X A^\top \\ A\Sigma_X & A\Sigma_X A^\top + \Sigma_\varepsilon \end{pmatrix} =: \begin{pmatrix} \Sigma_X & \Sigma_{XY} \\ \Sigma_{XY} & \Sigma_Y \end{pmatrix}. \tag{48}$$

This can be derived using $\Sigma_{XY} \equiv \text{Cov}(X, Y) = \mathbb{E}[XY^\top] - \mathbb{E}[X]\mathbb{E}[Y]^\top$ etc. The mutual information between $X$ and $Y$ is (cf. Eq. (22))

$$I(X; Y) = \frac{1}{2} \log\left( \frac{|\Sigma_Y|}{|\Sigma_{Y|X}|} \right), \tag{49}$$

where $\Sigma_{Y|X} = \Sigma_\varepsilon$ is just the average covariance of the distributions $Y|X \sim x = \mathcal{N}(Ax, \Sigma_\varepsilon)$, and $\Sigma_Y$ is defined in Eq. (48). If we use a VP process $p_{\text{eq}} = \mathcal{N}(0, I)$, and at sufficiently large $T$ (cf. Eq. (3)),

$$S_{\text{tot}}^X \approx D_{\text{KL}}(p_\text{d}\|p_{\text{eq}}) = D_{\text{KL}}\left( \mathcal{N}(0, \Sigma_X)\|\mathcal{N}(0, I) \right) = \frac{1}{2}\left( \text{Tr}(\Sigma_X) - \log|\Sigma_X| - D_X \right). \tag{50}$$

*Determinism:* In one set of experiments we set $A \sim \mathcal{N}(0, 1)^{D_X \times D_Y}$, $\Sigma_\varepsilon = \sigma_\varepsilon^2 I$, and

$$\Sigma_X = HH^\top + \epsilon I, \tag{51}$$

where $H \sim \mathcal{N}(0, 1/D_X)^{D_X \times D_X}$ and a small $\epsilon > 0$ ensures numerical stability as well as positive definiteness of $\Sigma_X$. The $\Sigma_X$ from Eq. (51) is full-rank, so $S_{\text{tot}}^X$ is well-behaved. We vary $\sigma_\varepsilon$ to adjust the correlation between $X$ and $Y$, with $I(X; Y)$ growing large as $Y$ becomes a more deterministic function of $X$ at small $\sigma_\varepsilon$ (see Fig. 12).

*Flattening:* In another set of experiments, we simulate the manifold problem in $X$ by setting

$$\Sigma_X = U\Lambda U^\top + \epsilon I, \quad \Lambda = \text{diag}(\lambda_1, \cdots, \lambda_k, \overbrace{\lambda_\delta, \cdots, \lambda_\delta}^{D_X - k}), \tag{52}$$

where $\{\lambda_i\}_{i=1}^k \sim |\mathcal{N}(0, 1/k)| + 0.1$, and with smaller values of $\lambda_\delta$ effectively lowering the rank of $X$. This time $\sigma_\varepsilon$ is kept sufficiently large that $I(X; Y)$ remains finite even as $S_{\text{tot}}^X$ rapidly.

We can also compute the ideal score functions $\nabla \log p(\tilde{x}_s, s|y)$ and $\nabla \log p(\tilde{x}_s, s)$ under the forward process. The conditional density at time $s$ are obtained by evolving the initial distribution of $X|Y \sim y$, namely (see Bishop, 2006, Sec. 2.3.1)

$$\mathcal{N}\left( \Sigma_{XY}\Sigma_Y^{-1}y, \Sigma_X - \Sigma_{XY}\Sigma_Y^{-1}\Sigma_{XY} \right) =: \mathcal{N}\left( \mu_{X|Y}, \Sigma_{X|Y} \right). \tag{53}$$

We use the VP process in our experiments, under which

$$\tilde{x}_s = \sqrt{\alpha(s)}x + \sqrt{1 - \alpha(s)}\eta, \quad \eta \sim \mathcal{N}(0, I), \tag{54}$$

where $\alpha(s) = \exp\left( -\int_0^s d\bar{s}\beta(\bar{s}) \right)$. Therefore, Eq. (53) is diffused to

$$p(\tilde{x}_s, s|y) = \mathcal{N}(\tilde{x}_s; \mu_s^{X|Y}, \Sigma_s^{X|Y}), \tag{55}$$

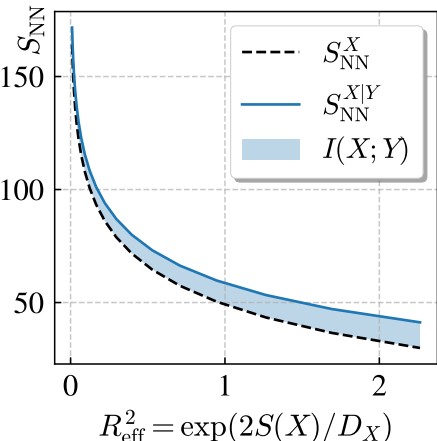

*Figure 9.* Neural entropy and mutual information in a joint Gaussian model $\boldsymbol{Y} = A\boldsymbol{X} + \boldsymbol{\varepsilon}$ (see Sec. D.1). $I(\boldsymbol{X};\boldsymbol{Y})$ becomes a vanishingly smaller fraction of the neural entropy $S_{\mathrm{NN}}^{\boldsymbol{X}}$ as $R_{\mathrm{eff}}$, the effective size of $p_{\mathrm{d}}(\boldsymbol{x})$, is lowered by decreasing $\lambda_{\delta}$.

$$\mu_s^{\boldsymbol{X}|\boldsymbol{Y}} := \mathbb{E}[\tilde{\boldsymbol{x}}_s|\boldsymbol{y}] = \sqrt{\alpha(s)}\Sigma_{\boldsymbol{X}\boldsymbol{Y}}\Sigma_{\boldsymbol{Y}}^{-1}\boldsymbol{y} \equiv \sqrt{\alpha(s)}\mu_{\boldsymbol{X}|\boldsymbol{Y}},$$

$$\begin{aligned}
\Sigma_s^{\boldsymbol{X}|\boldsymbol{Y}} :=&\, \mathrm{Cov}[\boldsymbol{X}_s|\boldsymbol{y}, \boldsymbol{X}_s|\boldsymbol{y}] \\
=&\, \mathbb{E}[(\boldsymbol{X}_s - \mu_s^{\boldsymbol{X}|\boldsymbol{Y}})(\boldsymbol{X}_s - \mu_s^{\boldsymbol{X}|\boldsymbol{Y}})^{\top}] \\
=&\, \alpha(s)\mathbb{E}[(\boldsymbol{X} - \mu_{\boldsymbol{X}|\boldsymbol{Y}})(\boldsymbol{X} - \mu_{\boldsymbol{X}|\boldsymbol{Y}})^{\top}] + (1 - \alpha(s))I \\
=&\, \alpha(s)\Sigma_{\boldsymbol{X}|\boldsymbol{Y}} + (1 - \alpha(s))I.
\end{aligned}$$

Under Eq. (54), the marginal density at $s$ is

$$p(\tilde{\boldsymbol{x}}_s, s) = \mathcal{N}(\tilde{\boldsymbol{x}}_s; 0, \Sigma_s^{\boldsymbol{X}}), \quad \Sigma_s^{\boldsymbol{X}} = \alpha(s)\Sigma_{\boldsymbol{X}} + (1 - \alpha(s))I. \tag{56}$$

Similarly, if the joint distribution were evolved by a VP process acting on both components of $\boldsymbol{R}$, the density at an intermediate time is

$$p(\tilde{\boldsymbol{x}}_s, \tilde{\boldsymbol{y}}_s, s) =: p(\tilde{\boldsymbol{r}}_s, s) = \mathcal{N}(\tilde{\boldsymbol{r}}_s; 0, \Sigma_s), \quad \Sigma_s^{\boldsymbol{R}} = \alpha(s)\Sigma_{\boldsymbol{R}} + (1 - \alpha(s))I. \tag{57}$$

Then, the conditional, marginal, and joint scores are

$$\nabla_{\tilde{\boldsymbol{x}}_s} \log p(\tilde{\boldsymbol{x}}_s, s|\boldsymbol{y}) = -\left(\Sigma_s^{\boldsymbol{X}|\boldsymbol{Y}}\right)^{-1}\left(\tilde{\boldsymbol{x}}_s - \mu_s^{\boldsymbol{X}|\boldsymbol{Y}}\right), \tag{58a}$$

$$\nabla_{\tilde{\boldsymbol{x}}_s} \log p(\tilde{\boldsymbol{x}}_s, s) = -\left(\Sigma_s^{\boldsymbol{X}}\right)^{-1}\tilde{\boldsymbol{x}}_s, \tag{58b}$$

$$\nabla_{\tilde{\boldsymbol{r}}_s} \log p(\tilde{\boldsymbol{r}}_s, s) = -\left(\Sigma_s^{\boldsymbol{R}}\right)^{-1}\tilde{\boldsymbol{r}}_s. \tag{58c}$$

Equipped with these formulas, we can verify points 1 to 3 while sidestepping the singular behavior of neural entropy in image diffusion models. For each experiment with the joint Gaussian, a conditional diffusion model is trained on $\{(\boldsymbol{x}^{(i)}, \boldsymbol{y}^{(i)})\}_{i=1}^{N}$, and a second model learns the marginal of $\boldsymbol{X}$ from $\{(\boldsymbol{x}^{(i)})\}_{i=1}^{N}$ alone. These models use a simple MLP core, and we train with the maximum likelihood denoising objective (Eq. (40) with $\lambda(s) = \sigma(s)^2/2$). The experiments are described below.

**Entropy and correlation**  We train on samples of Eq. (47) for $D_{\boldsymbol{X}} = 25$, $D_{\boldsymbol{Y}} = 15$, with $A$ kept fixed and $\sigma_{\boldsymbol{\varepsilon}} = 1.0, 0.6, 0.25$. Reducing the noise strength increases $I(\boldsymbol{X};\boldsymbol{Y})$ (cf. Fig. 6), as well as the conditional neural entropy $S_{\mathrm{NN}}^{\boldsymbol{X}|\boldsymbol{Y}}$. We also plot the true value of these quantities calculated with the analytic scores in Eq. (58c). The resulting entropy curves are shown in Fig. 12. Notice how the peak of the entropy rate curves becomes more localized at earlier $s$ as the correlation between $\boldsymbol{X}$ and $\boldsymbol{Y}$ is made stronger. This is the same effect that gives rise to the sharp peak in the neural entropy rates for image diffusion models (see Fig. 16).

**Mutual information and guidance** In Sec. 3, we explained how CFG increases $I(\boldsymbol{X}; \boldsymbol{Y})$. For the joint Gaussian, $\boldsymbol{Y}$ is not a discrete random variable like a class label. Nonetheless, we can study how a 'CFG-style' modification to the reverse drift affects the samples from Eq. (47). Since we know the true scores, we can produce samples with the probability flow ODE (Maoutsa et al., 2020),

$$\mathrm{d}\boldsymbol{x}_t = \left( -b_+(\boldsymbol{x}_t, T-t) + \frac{\sigma(T-t)^2}{2} \left[ (1+w)\nabla \log p(\boldsymbol{x}_t, t|\boldsymbol{y}) - w\nabla \log p(\boldsymbol{x}_t, t) \right] \right) \mathrm{d}t. \tag{59}$$

A simple example of the samples generated by Eq. (59) is shown in Fig. 8. CFG tightens the dependence of $\boldsymbol{X}$ on $\boldsymbol{Y}$, but also skews the true relationship between them (see Sec. C.2). Going to higher dimensions, we set $D_{\boldsymbol{X}} = 25$ and generate training data with Eq. (59) for $D_{\boldsymbol{Y}} = 5, 10, 25$, with a range of CFG weights $w \in (0, 6)$. We train a pair of diffusion models to reconstruct $\boldsymbol{X}|\boldsymbol{Y}$ and $\boldsymbol{X}$, and estimate $I(\boldsymbol{X}; \boldsymbol{Y})$ using the MINDE formula, Eq. (13). The results are plotted in Fig. 10.

As expected, CFG does increase the mutual information between $\boldsymbol{X}$ and $\boldsymbol{Y}$. Two observations stand out: first, the increase in $I(\boldsymbol{X}; \boldsymbol{Y})$ saturates at larger $w$, and second, the gain in $I(\boldsymbol{X}; \boldsymbol{Y})$ is higher at larger $D_{\boldsymbol{Y}}$. Both these features can be understood through the information bottleneck principle from Sec. A.2: the degree to which the binding between $\boldsymbol{X}$ and $\boldsymbol{Y}$ can be strengthened is limited by the number of degrees of freedom in $\boldsymbol{Y}$.

**Analytic formula** The linear model from Eq. (47) is special in that the CFG-modified score is also affine in $\tilde{\boldsymbol{x}}_s$, which means

$$(1+w)\nabla \log p(\tilde{\boldsymbol{x}}_s, s|\boldsymbol{y}) - w\nabla \log p(\tilde{\boldsymbol{x}}_s, s) =: \nabla \log \hat{p}(\tilde{\boldsymbol{x}}_s, s|\boldsymbol{y}), \tag{60}$$

where $\hat{p}$ is a Gaussian,

$$\hat{p}(\tilde{\boldsymbol{x}}_s, s|\boldsymbol{y}) = \mathcal{N}(\tilde{\boldsymbol{x}}_s; \hat{\mu}_s, \hat{\Sigma}_s), \tag{61a}$$

$$\hat{\Sigma}_s^{-1} := (1+w)\left(\Sigma_s^{\boldsymbol{X}|\boldsymbol{Y}}\right)^{-1} - w\left(\Sigma_s^{\boldsymbol{X}}\right)^{-1} \tag{61b}$$

$$\hat{\mu}_s := (1+w)\hat{\Sigma}_s\left(\Sigma_s^{\boldsymbol{X}|\boldsymbol{Y}}\right)^{-1}\mu_s^{\boldsymbol{X}|\boldsymbol{Y}}. \tag{61c}$$

At $s = 0$ we obtain the target distribution generated by the reverse process (cf. Eq. (53)),

$$\hat{p}_{\mathrm{d}}(\boldsymbol{x}|\boldsymbol{y}) = \mathcal{N}(\boldsymbol{x}; \hat{\mu}, \hat{\Sigma}), \tag{62a}$$

$$\hat{\Sigma}^{-1} := \Sigma_{\boldsymbol{X}|\boldsymbol{Y}}^{-1} + w\left(\Sigma_{\boldsymbol{X}|\boldsymbol{Y}}^{-1} - \Sigma_{\boldsymbol{X}}^{-1}\right), \tag{62b}$$

$$\hat{\mu} := (1+w)\hat{\Sigma}\Sigma_{\boldsymbol{X}|\boldsymbol{Y}}^{-1}\Sigma_{\boldsymbol{X}\boldsymbol{Y}}\Sigma_{\boldsymbol{Y}}^{-1}\boldsymbol{y} =: \mathfrak{M}\boldsymbol{y}. \tag{62c}$$

For $w \geq 0$, the distribution $\hat{p}_{\mathrm{d}}$ is well-defined, since $\Sigma_{\boldsymbol{X}|\boldsymbol{Y}}^{-1} - \Sigma_{\boldsymbol{X}}^{-1}$ is positive semidefinite. For $w > 0$ the covariance contracts along directions where $\Sigma_{\boldsymbol{X}|\boldsymbol{Y}}$ differs most from $\Sigma_{\boldsymbol{X}}$. The resulting $\hat{p}_{\mathrm{d}}$ is Gaussian, but it is *not* a conditional of the true joint Gaussian $p_{\mathrm{d}}(\boldsymbol{x}, \boldsymbol{y})$. Rather, it is the conditional of the joint density $\hat{p}_{\mathrm{d}}(\boldsymbol{x}, \boldsymbol{y}) = \hat{p}_{\mathrm{d}}(\boldsymbol{x}|\boldsymbol{y})p_{\mathrm{d}}(\boldsymbol{y})$, where $p_{\mathrm{d}}(\boldsymbol{y}) \equiv \mathcal{N}(0, \Sigma_{\boldsymbol{Y}})$ is the true marginal of $\boldsymbol{y}$.[3] Then, we can compute $I(\boldsymbol{X}; \boldsymbol{Y})_{\mathrm{CFG}}$ by first drawing $\boldsymbol{y}$ from the true marginal $p_{\mathrm{d}}(\boldsymbol{y})$, and then drawing $\boldsymbol{x}|\boldsymbol{y}$ from the CFG-generated conditional $\hat{p}_{\mathrm{d}}(\boldsymbol{x}|\boldsymbol{y})$. This is the mutual information measured in the solid curves in Fig. 10. We can also derive an analytic expression for $I(\boldsymbol{X}; \boldsymbol{Y})_{\mathrm{CFG}}$,[4] using the fact that the mutual information for *any* joint Gaussian pair $(\boldsymbol{X}, \boldsymbol{Y})$ is (cf. Eq. (49))

$$I(\boldsymbol{X}; \boldsymbol{Y}) = \frac{1}{2}\log \frac{|\mathrm{Cov}(\boldsymbol{X})|}{|\mathrm{Cov}(\boldsymbol{X}|\boldsymbol{Y})|}. \tag{63}$$

In the CFG case, $\mathrm{Cov}(\boldsymbol{X}|\boldsymbol{Y}) = \hat{\Sigma}$. The marginal covariance can be computed using the law of total covariance,

$$\mathrm{Cov}(\boldsymbol{X}) = \mathbb{E}_{\boldsymbol{y}}[\mathrm{Cov}(\boldsymbol{X}|\boldsymbol{y})] + \mathrm{Cov}_{\boldsymbol{y}}(\mathbb{E}[\boldsymbol{X}|\boldsymbol{y}]). \tag{64}$$

---

[3]Reverse evolution with the CFG drift visits a family of intermediate densities that no forward diffusion produces, consistent with the discussion in Sec. C.2. For instance, the VP process applied to Eq. (62c) produces moments different from those in Eq. (61c).

[4]We thank ICML reviewer jYnY for pointing this out.

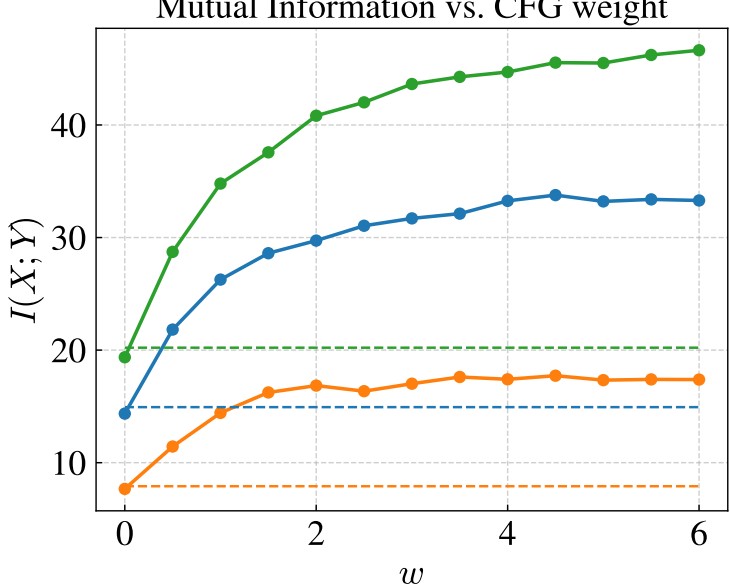

*Figure 10.* Mutual information under CFG for the joint Gaussian model from Eq. (47). We fix $D_X = 25$ and repeat the experiment with $D_Y = 5, 10, 15$. Notice how $I(X;Y)_{\text{CFG}}$ increases as the guidance strength is ramped up. It saturates faster for smaller $D_Y$, when $Y$ has fewer degrees of freedom to encode the diversity in $X$. This is also why the mutual information between images and labels is low in the first place. $I(X;Y)_{\text{CFG}}$ was estimated using Eq. (13), with diffusion models trained on data generated with CFG. The true value of mutual information is known from Eq. (49).

The first term is simply $\hat{\Sigma}$ since the covariance of $\hat{p}_\text{d}(x|y)$ is independent of $y$. The second term is

$$\text{Cov}_{\boldsymbol{y}}(\mathbb{E}[\boldsymbol{X}|\boldsymbol{y}]) \overset{(62c)}{=} \text{Cov}_{\boldsymbol{y}}(\mathfrak{M}\boldsymbol{y}) = \mathbb{E}[(\mathfrak{M}\boldsymbol{y})(\mathfrak{M}\boldsymbol{y})^\top] = \mathfrak{M}\Sigma_{\boldsymbol{Y}}\mathfrak{M}^\top. \tag{65}$$

Putting it all together,

$$I(\boldsymbol{X};\boldsymbol{Y})_{\text{CFG}} = \frac{1}{2}\log\frac{|\hat{\Sigma} + \mathfrak{M}\Sigma_{\boldsymbol{Y}}\mathfrak{M}^\top|}{|\hat{\Sigma}|} = \frac{1}{2}\log|I_{D_{\boldsymbol{X}}} + \hat{\Sigma}^{-1}\mathfrak{M}\Sigma_{\boldsymbol{Y}}\mathfrak{M}^\top|. \tag{66}$$

Focusing on the $\hat{\Sigma}^{-1}\mathfrak{M}\Sigma_{\boldsymbol{Y}}\mathfrak{M}^\top$ term, we see that there is an overall factor of $(1 + w)^2$ that comes from the $\mathfrak{M}$'s (cf. Eq. (62c)), which scales as $\approx w^2$ for large $w$. In the same limit, a $\hat{\Sigma} \propto w^{-1}$ (cf. Eq. (62b)) is sourced from the $\mathfrak{M}^\top$ piece, but not $\hat{\Sigma}^{-1}\mathfrak{M}$. Furthermore, $\hat{\Sigma}^{-1}\mathfrak{M}\Sigma_{\boldsymbol{Y}}\mathfrak{M}^\top$ has a rank of at most $D_{\boldsymbol{Y}}$. Thus $I(\boldsymbol{X};\boldsymbol{Y})_{\text{CFG}} \propto D_{\boldsymbol{Y}}\log w$, which explains how mutual information saturates with guidance strength and $D_{\boldsymbol{Y}}$ in Fig. 10.

## E. Information Hierarchy

### E.1. Diffusion Autoencoders

A diffusion model can develop its own side information when it is paired with an encoder. This arrangement is called a *diffusion autoencoder*, or DAE (Preechakul et al., 2022). Recall that a standard variational autoencoder (VAE) is trained to minimize the negative of the evidence lower bound (Kingma & Welling, 2014),

$$-\text{ELBO}(\boldsymbol{x}) = \mathbb{E}_{q_\phi(\boldsymbol{z}|\boldsymbol{x})}[-\log p_\theta(\boldsymbol{x}|\boldsymbol{z})] + \gamma D_{\text{KL}}(q_\phi(\boldsymbol{z}|\boldsymbol{x}) \| p(\boldsymbol{z})) \tag{67}$$

where $\phi$ and $\theta$ are the encoder and decoder parameters, respectively. The coefficient $\gamma$ plays the role of the weighting factor in the $\beta$-VAE objective (Higgins et al., 2017), balancing reconstruction and KL terms. In a DAE, the reconstruction

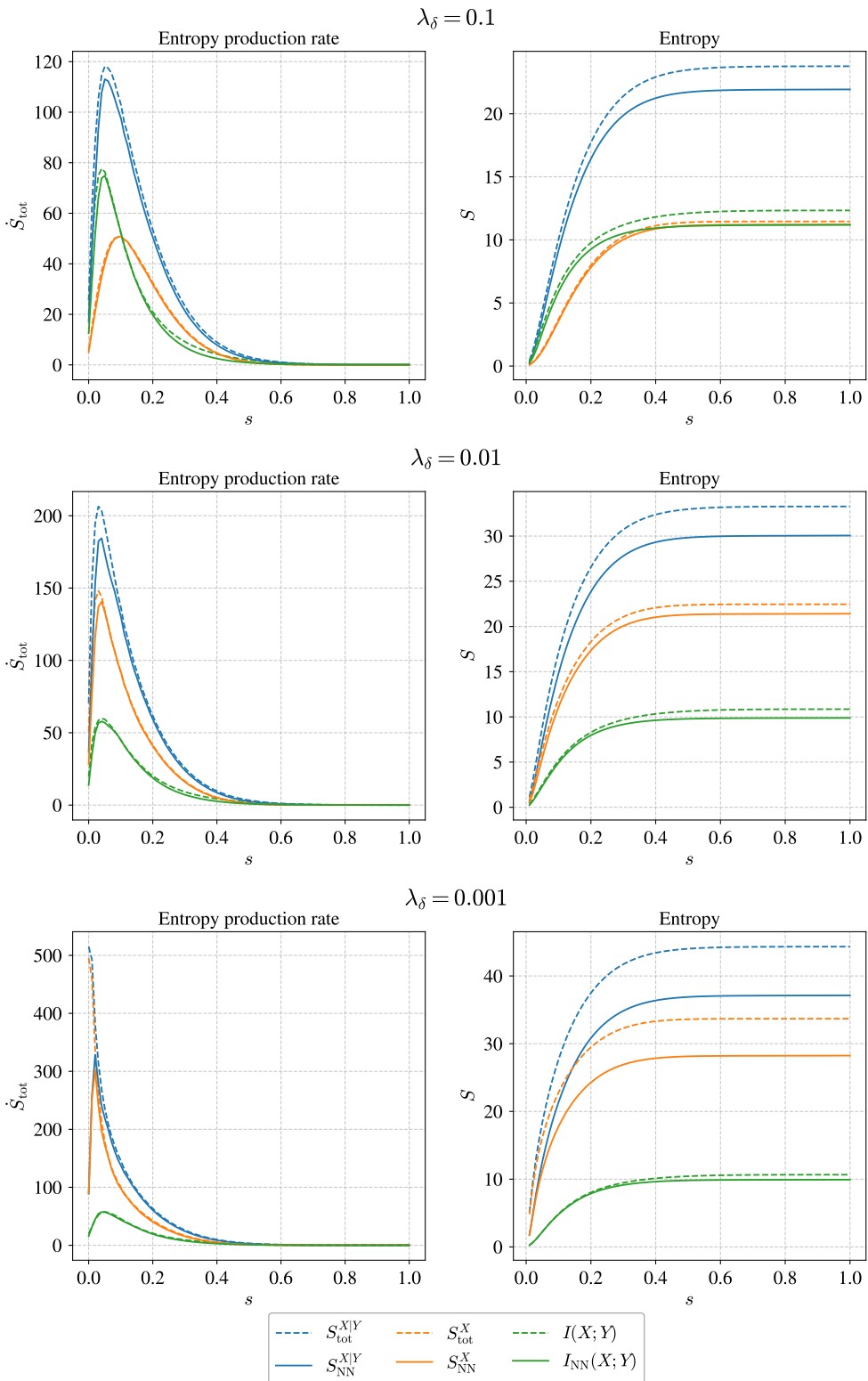

*Figure 11.* *Flattening:* Evolution of total entropy, neural entropy, and the mutual information under the forward process, for a joint Gaussian with $D_{\boldsymbol{X}} = 25, D_{\boldsymbol{Y}} = 15$, as the effective rank of $\Sigma_{\boldsymbol{X}}$ is made smaller. We keep $D_{\boldsymbol{X}} = 25, D_{\boldsymbol{Y}} = 15, \sigma_{\boldsymbol{\varepsilon}} = 1.0$ but $\lambda_\delta$ is lowered from top to bottom. This makes $S_{\mathrm{tot}}^{\boldsymbol{X}}$ grow rapidly while $I(\boldsymbol{X}; \boldsymbol{Y})$ remains finite. The bottom plot shows the diffusion model struggling to absorb information at the early time steps when $S_{\mathrm{tot}}^{\boldsymbol{X}}$ and $S_{\mathrm{tot}}^{\boldsymbol{X}|\boldsymbol{Y}}$ are both large. Despite this, the models compute $I(\boldsymbol{X}; \boldsymbol{Y})$ more or less correctly. See Sec. 5 for discussion.

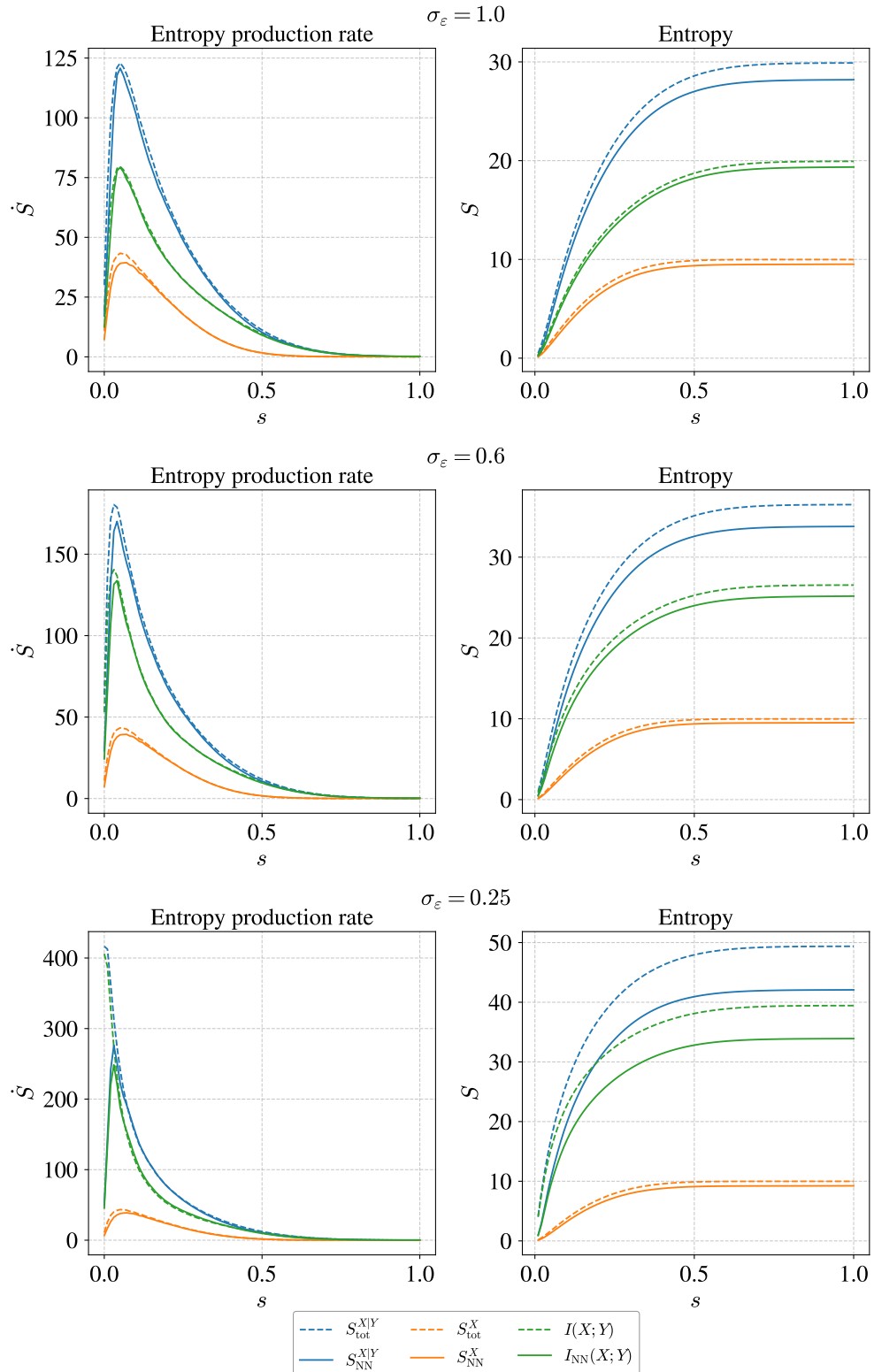

*Figure 12.* **Determinism:** Entropy curves for a joint Gaussian with $D_{\boldsymbol{X}} = 25, D_{\boldsymbol{Y}} = 15$ and a full-rank $\Sigma_{\boldsymbol{X}}$. As $\sigma_{\boldsymbol{\varepsilon}}$ is lowered $\boldsymbol{X}$ and $\boldsymbol{Y}$ become more correlated, which causes $I(\boldsymbol{X}; \boldsymbol{Y})$ to grow while $S_{\text{tot}}^{\text{X}}$ remain fixed. Notice also how the $S_{\text{tot}}^{\boldsymbol{X}|\boldsymbol{Y}}$ and $I(\boldsymbol{X}; \boldsymbol{Y})$ curves become more concentrated near $s = 0$; as $\sigma_{\boldsymbol{\varepsilon}} \to 0$, $\boldsymbol{X}$ and $\boldsymbol{Y}$ converge on the hyperplane $\boldsymbol{y} = A\boldsymbol{x}$ which takes an infinite amount of information to locate precisely in the joint $\boldsymbol{xy}$ space, even though $\boldsymbol{X}$ itself is not lower-dimensional. The diffusion model no longer captures $I(\boldsymbol{X}; \boldsymbol{Y})$ accurately.

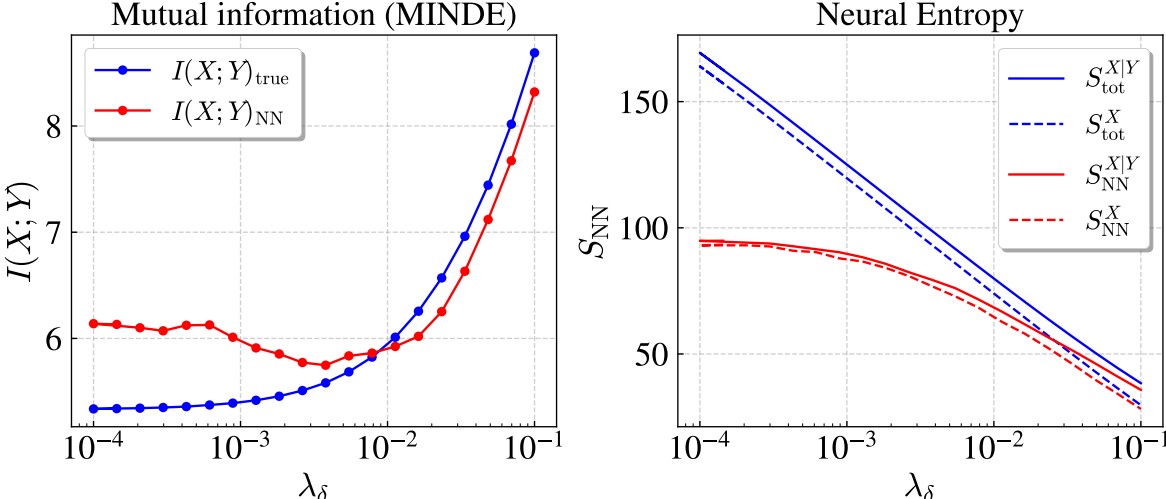

*Figure 13.* Mutual information and neural entropy for the joint Gaussian model as $\boldsymbol{X}$ is flattened. We used $D_{\boldsymbol{X}} = 50, D_{\boldsymbol{Y}} = 8, k = 10$ and $\sigma_{\boldsymbol{\varepsilon}} = 1$. The blue lines are computed with the analytic expressions Eqs. (49) and (50). The variable $\lambda_\delta$ controls the spread of $\boldsymbol{X}$ in $D_{\boldsymbol{X}} - k$ directions (cf. Eq. (52)). It is made smaller while $\boldsymbol{Y}$ is maintained at full rank (see Fig. 11), which makes $S_{\text{tot}}^{\boldsymbol{X}}$ grow rapidly. The network struggles to keep pace at low $\lambda_\delta$, as evident in the divergence between the $S_{\text{tot}}$ and $S_{\text{NN}}$. In this range, even the $I(\boldsymbol{X}; \boldsymbol{Y})$ estimate begins to suffer.

term is replaced by the upper bound

$$\mathbb{E}_{\boldsymbol{X},\boldsymbol{Z}}[-\log p_{\boldsymbol{\theta}}(\boldsymbol{x}|\boldsymbol{z})] + c \leq \tag{68}$$
$$\int_0^T \mathrm{d}s\, \mathbb{E}_{\boldsymbol{X},\boldsymbol{Z},\tilde{\boldsymbol{X}}_s} \left[ \frac{\sigma^2}{2} \left\| \nabla \log p_{\text{eq}}^{(s)}(\tilde{\boldsymbol{x}}_s) - \nabla \log p(\tilde{\boldsymbol{x}}_s, s|\boldsymbol{x}, 0) + \boldsymbol{e}_{\boldsymbol{\theta}}(\tilde{\boldsymbol{x}}_s, s; \boldsymbol{z}) \right\|^2 \right] =: \mathcal{L}_{\text{DEM}}^{\boldsymbol{X}|\boldsymbol{Z}}.$$

Here $c$ is a constant with respect to the network parameters $\boldsymbol{\theta}$ (cf. Eq. (30)). The expectation over $\boldsymbol{X}$ and $\tilde{\boldsymbol{X}}_s$ averages over the data points $\{\boldsymbol{x}^{(i)}\}_{i=1}^N$ and their value at time $s$ under the forward process in Eq. (1). Importantly, the bound is precisely the denoising entropy-matching objective used to train the diffusion model; minimizing this loss is equivalent to maximizing log likelihood (see Sec. B.1). A score-matching parameterization can also be used in Eq. (68). The latents $\boldsymbol{z}$ are sampled from the encoder,

$$q_{\boldsymbol{\phi}}(\boldsymbol{z}|\boldsymbol{x}) = \mathcal{N}(\boldsymbol{z}; \mu_{\boldsymbol{\phi}}(\boldsymbol{x}), \text{diag}(\sigma_{\boldsymbol{\phi}}^2(\boldsymbol{x}))), \tag{69}$$

using the reparameterization trick to enable gradient-based training. Conditioning on $\boldsymbol{z}$ allows the diffusion model to concentrate the probability mass to a smaller region in $\boldsymbol{x}$-space compared to the unconditional case; on average, conditional distributions are narrower than the marginals (cf. Eq. (10)). If the diffusion model had perfect freedom to choose the latent, it would assign a unique $\boldsymbol{z}^{(i)}$ to each $\boldsymbol{x}^{(i)}$ in the dataset, since that would lead to maximal concentration of probability in each conditional distribution. However, the DAE is unable to do so because (i) the inductive biases of the diffusion model temper its ability to perfectly resolve each $\boldsymbol{x}^{(i)}$, which is desirable because it avoids overfitting (Kadkhodaie et al., 2023), and (ii) the encoder admits a narrow range of $\boldsymbol{z}$, so the diffusion decoder has a limited set of latent codes to choose from—the DAE is an *information bottleneck* (see Sec. A.2). Therefore, jointly minimizing the encoder term with the upper bound from Eq. (68) forces the diffusion model to negotiate a latent $\boldsymbol{Z}$ that is maximally correlated with $\boldsymbol{X}$, under the given constraints. This follows from

$$\max I(\boldsymbol{X}; \boldsymbol{Z}) \equiv S(\boldsymbol{X}) - \min S(\boldsymbol{X}|\boldsymbol{Z}) \equiv S(\boldsymbol{X}) - \min \mathbb{E}_{\boldsymbol{X},\boldsymbol{Z}}[-\log p_{\boldsymbol{\theta}}(\boldsymbol{x}|\boldsymbol{z})], \tag{70}$$

since the cross entropy $\mathbb{E}_{\boldsymbol{X},\boldsymbol{Z}}[-\log p_{\boldsymbol{\theta}}(\boldsymbol{x}|\boldsymbol{z})]$ upper bounds the conditional entropy $S(\boldsymbol{X}|\boldsymbol{Z})$, and $S(\boldsymbol{X}) := \mathbb{E}_{\boldsymbol{X}}[-\log p_{\text{d}}(\boldsymbol{x})]$ is independent of $\boldsymbol{\theta}$ or $\boldsymbol{\phi}$. The latent $\boldsymbol{Z}$ is a compressed proxy for how the diffusion model represents $\boldsymbol{X}$. We use this fact in Sec. E, where the hierarchical nature of the information stored in these models is revealed through the structure they induce on the latents.

Minimizing Eq. (68) implicitly *maximizes* $S_{\text{NN}}^{\boldsymbol{X}|\boldsymbol{Z}}$, as evident from Eqs. (9) and (70)—a strongly correlated latent forces the diffusion model to discern tighter (on average) distributions of $\boldsymbol{X}|\boldsymbol{Z}$, which requires a higher neural entropy, whereas

Top: Original Images | Middle: DAE Reconstructions | Bottom: VAE Reconstructions

*Figure 14.* Images reconstructed by a DAE and VAE. Both of them have the same encoder architecture. The VAE uses a Gaussian decoder that tends to produce blurrier outputs, whereas the diffusion decoder captures significantly more textural detail, leading to sharper images. In this example, the convolutional encoder's simplicity limits the fidelity of the DAE reconstruction.

a weak latent does the opposite. This makes the DAE a great conceptual tool to understand how conditioning affects retention. Consider first the limiting case where the encoder is just the identity operator, so $\boldsymbol{Z} = \boldsymbol{X}$. There is now a unique $\boldsymbol{z}^{(i)}$ for each $\boldsymbol{x}^{(i)}$, so every $p_{\boldsymbol{\theta}}(\boldsymbol{x}|\boldsymbol{z})$ is a delta function, and the neural entropy $S_{\mathrm{NN}}^{\boldsymbol{x}|\boldsymbol{z}}$ is very large—the diffusion model has *memorized* each $\boldsymbol{x}^{(i)}$. At the other extreme, we can imagine an encoder that maps every value of $\boldsymbol{x}^{(i)}$ to a single value, call it $\boldsymbol{z}_{\mathrm{null}}$. This converts the decoder into an unconditional diffusion model since it receives no information about $\boldsymbol{X}$ from the encoder. Consequently, the model learns to reconstruct the broadest possible distribution of $\boldsymbol{X}$, and $S_{\mathrm{NN}}^{\boldsymbol{x}|\boldsymbol{z}}$ reaches its lowest possible value, $S_{\mathrm{NN}}^{\boldsymbol{x}}$. So the model retains the smallest amount of information when it is least committed to recovering each $\boldsymbol{x}^{(i)}$ perfectly.

This argument also connects to the tension between conditioning and generalization. In Sec. 3 we discussed the weak correlation between images $\boldsymbol{X}$ and their labels $\boldsymbol{Y}$. If $I(\boldsymbol{X};\boldsymbol{Y})$ were stronger, it would reduce the diversity in samples produced because the model has memorized more information. The power of CFG is that it is applied during the generative stage, so the model does not have to overcommit to the given data during training. However, CFG does have a fundamental limitation: if the underlying dependence between $\boldsymbol{X}$ and $\boldsymbol{Y}$ is weak, amplification of the signal can only go so far. DAE's allow an alternative approach: a second diffusion model is trained to generate from $\boldsymbol{Y}$ the latent $\boldsymbol{Z}$ first, which is then used to produce $\boldsymbol{X}$. The latent $\boldsymbol{Z}$ abstracts away the perceptual details that overwhelm the correlation between $\boldsymbol{X}$ and $\boldsymbol{Y}$, while also being expressive enough to encode the variation in the semantic structure of $\boldsymbol{X}$ (Preechakul et al., 2022).

### E.2. VAE vs. DAE

In Sec. E.1 we discussed diffusion autoencoders and pointed out that they help understand how the diffusion models store information. To see how this works, we start by comparing the diffusion model in the DAE with a simpler Gaussian-likelihood decoder, $p_{\boldsymbol{\psi}}(\boldsymbol{x}|\boldsymbol{z}) = \mathcal{N}(\boldsymbol{x}; f_{\boldsymbol{\psi}}(\boldsymbol{z}), \sigma_{\mathrm{dec}}^2 I)$, where $\sigma_{\mathrm{dec}}$ is a constant and $\boldsymbol{\psi}$ are the network parameters. Minimizing the $\ell_2$ loss of this decoder,

$$\mathbb{E}_{q_{\boldsymbol{\phi}}(\boldsymbol{z}|\boldsymbol{x})}[-\log p_{\boldsymbol{\psi}}(\boldsymbol{x}|\boldsymbol{z})] \propto \mathbb{E}_{\boldsymbol{z} \sim q_{\boldsymbol{\phi}}(\cdot|\boldsymbol{x})}[\|\boldsymbol{x} - f_{\boldsymbol{\psi}}(\boldsymbol{z})\|^2], \tag{71}$$

is equivalent to predicting the the conditional mean $\mathbb{E}[\boldsymbol{x}|\boldsymbol{z}]$, which lies between the modes of the true distribution (Bishop, 2006). As a result, in image processing applications, the reconstructions from such a decoder tend to be blurry (Wang & Bovik, 2009). On the other hand, the diffusion decoder from Eq. (68) generates a new sample by progressively evolving a random vector toward a denoised mean that becomes more resolved over time. Therefore, these models can capture the multi-modal structure of the underlying distribution with greater fidelity, producing reconstructions that are far more faithful to the original signal (see Fig. 14). Since the diffusion decoder retains more information about each $\boldsymbol{x}$, it can distinguish samples with greater accuracy. This places a greater strain on the encoder as it is pressured to supply more differentiated latent codes to disambiguate the wider variety of data points.

The latents in an autoencoder serve as a probe of the decoder's ability to capture information. This is borne out in a simple experiment comparing the latents from a DAE to those from a VAE with the Gaussian decoder in Eq. (71), both of which use the same encoder architecture. We train both autoencoders to reconstruct MNIST images, restricting ourselves to latent

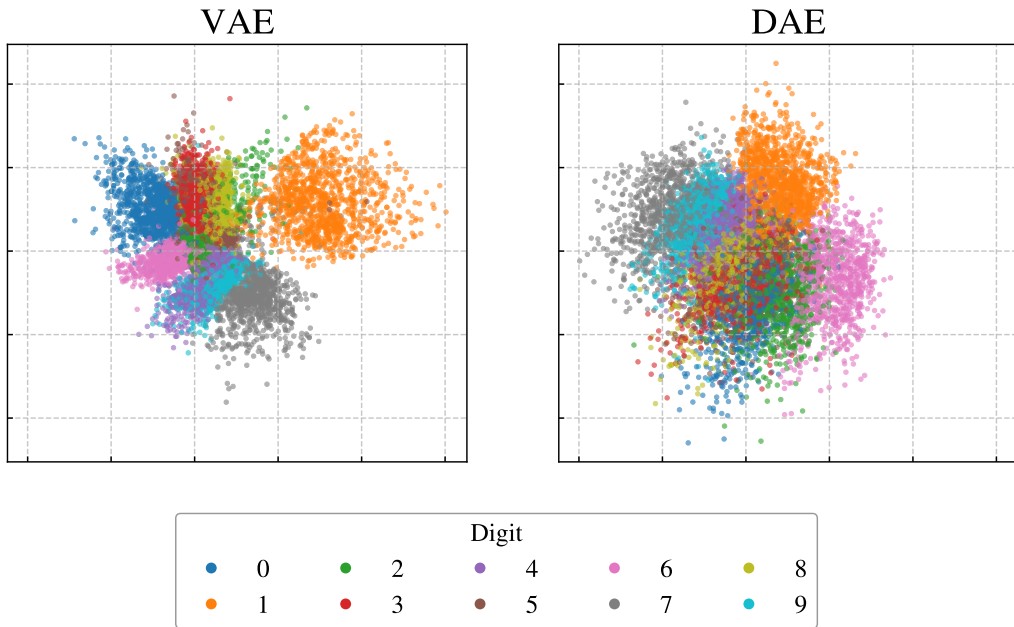

*Figure 15.* Latents from a VAE and DAE trained to reconstruct MNIST digits. Distinct clusters appear in the VAE latent, even at the low dimensionality of $D_Z = 2$. On the other hand, the DAE's latent clusters are more blended because the small-scale details captured by the diffusion decoder are similar for all digits, and this information overwhelms the semantic differences. See also Fig. 17a.

dimensions of $D_Z = 2$ for easier visualization of the aggregated posterior, $q_\phi(z) = \sum_x q_\phi(z|x)p_d(x)$. Even at this low $D_Z$, we observe discernible clustering in the VAE latent, corresponding to the different digits. By contrast, in the latent space of the DAE the clusters are more blended, with weaker separation between digit classes (see Fig. 15). This suggests that the DAE perceives greater similarity between different digits than the VAE, the common information across digit classes being the high-frequency detail washed away by the averaging effect of the Gaussian decoder. If we widen the bottleneck by increasing $D_Z$, we find better separation between the DAE clusters, since there is more room to encode the rich detail preserved by the diffusion decoder.

The above experiment gives us a clue as to why image diffusion models often neglect conditioning on class labels. The *semantic* information that identifies the digit '1' from an image of '1' is a relatively small fraction of the total information content in that image. The rest encodes *perceptual* details that have a similar distribution for all images, even those of different digits. Therefore, the marginal $X$ and the conditional $X \mid Y = y$ possess comparable entropy—specifying the class label does not reduce the uncertainty in $X$ by a lot. In other words, the mutual information between these images and their labels is low; the problem lies in the data itself. CFG is a trick to boost $I(X; Y)$ post-training, but it merely amplifies whatever signal is already present; *multiplying a weak signal also magnifies the noise*.

### E.3. Semantic vs. Perceptual

Why must the diffusion model devote a large fraction of its information budget to resolving the microscopic details of the image? And how do we know it is these details that overwhelm the semantic information? To answer these questions, we begin by noting that forward diffusion dissolves the perceptual details in the first few steps, whereas the semantic structure is preserved—we can still read off a digit from a noisy image of it. More prosaically, natural images follow a power-law spectrum, which means the low frequencies dominate while high frequency (short wavelength) modes are subdued (Ruderman, 1994; Dieleman, 2024a). Since the white noise term in Eq. (1) injects equal power across all frequencies, the finer details fade away more rapidly when images are diffused. Therefore, we expect entropy production associated with the removal of perceptual detail to be localized in a narrow interval near $s = 0$. Indeed, the neural entropy rates in Fig. 16 exhibit a sharp peak in this range, which answers the second question.

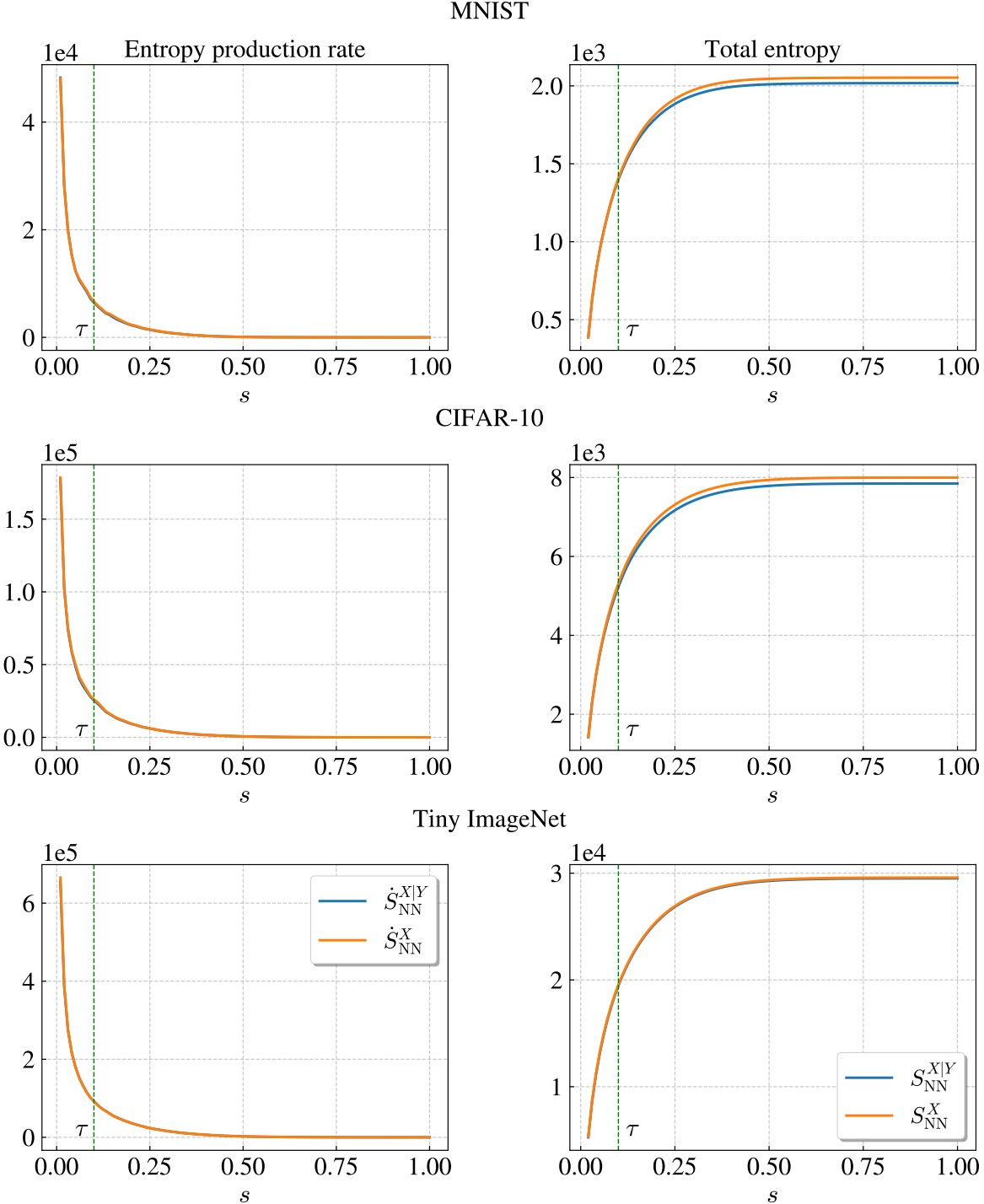

*Figure 16.* Neural entropy profiles for two image diffusion models trained on the MNIST, CIFAR-10, and Tiny ImageNet datasets. On the left is the entropy production rate, which is the time derivative of the neural entropy $S_{\mathrm{NN}}^{\bullet}$, defined in Eq. (11). The sharp rise in entropy rate at early $s$ is attributed to the low dimensionality of the data manifold. Notice how there is little difference between $S_{\mathrm{NN}}^{\boldsymbol{X}|\boldsymbol{Y}}$ and $S_{\mathrm{NN}}^{\boldsymbol{X}}$, because the latter is *much* larger than $I(\boldsymbol{X};\boldsymbol{Y})$ (cf. Eq. (10)). For MNIST, $S_{\mathrm{NN}}^{\boldsymbol{X}} = 2017.9$ and $I(\boldsymbol{X};\boldsymbol{Y}) = 4.1$ (from Eq. (12)). For CIFAR-10, $S_{\mathrm{NN}}^{\boldsymbol{X}} = 7776.2$ and $I(\boldsymbol{X};\boldsymbol{Y}) = 7.5404$. For Tiny ImageNet, $S_{\mathrm{NN}}^{\boldsymbol{X}} = 29586.2$ and $I(\boldsymbol{X};\boldsymbol{Y}) = 9.2868$. All measurements are in nats. Due to imperfections in learning, the values of $I(\boldsymbol{X};\boldsymbol{Y})$ are larger than their theoretical upper bound of $H(\boldsymbol{Y}) = \log 10 = 2.302$ nats for MNIST/CIFAR-10, and $H(\boldsymbol{Y}) = \log 200 = 5.928$ nats for Tiny ImageNet. But this ballpark estimate serves our main point about the smallness of conditioning information. The dashed red line indicates the partitioning of the denoising loss into semantic and perceptual pieces for the experiments in Sec. E.3.

We can also understand Fig. 16 from a geometric perspective by viewing $S_{\text{NN}}$ as the information the network injects in the $t$-direction. The data distribution resides on a low-dimensional submanifold of the ambient pixel space. The reduced dimensionality of the data manifold stems from the fact that nearby pixels are very strongly correlated in high-fidelity images, so there are fewer degrees of freedom than the naive pixel count (Lukoianov et al., 2025). In the generative stage, the diffusion model drives a high-dimensional Gaussian distribution back onto the lower-dimensional data manifold (see Fig. 7). The sharp rise in entropy rate as $t \to T$ is reflective of the fact that the network must supply substantial information to locate the manifold exactly, which involves collapsing the distribution to the low-dimensional data manifold $\mathcal{M}_{\boldsymbol{X}}$ (see Fig. 1). By the spectral argument above, the local details in the image are the last to be filled in, so $\mathcal{M}_{\boldsymbol{X}}$ is low dimensional because nearby pixels $x_k$ are nearly deterministic functions of others. Therefore, the singular behavior can also be traced back to the total correlation term in Eq. (7), as explained in Sec. 4, and at the end of Sec. A.1. This answers the first question.

We can use a DAE to test these claims. In a regular DAE, the latent $\boldsymbol{Z}$ compresses information from the entirety of the generative process that reconstructs $\boldsymbol{X}$. To probe the semantic (early $t$) and perceptual (late $t$) information separately, we can condition the denoising loss with two different latents $\boldsymbol{Z}_{\text{sem}}$ and $\boldsymbol{Z}_{\text{per}}$ in those intervals. Recall Eq. (68), which we shall write as

$$\mathbb{E}_{\boldsymbol{X},\boldsymbol{Z}}[-\log p_{\boldsymbol{\theta}}(\boldsymbol{x}|\boldsymbol{z})] + h(0,T) - \mathbb{E}_{\tilde{\boldsymbol{X}}_T}[-\log p_0] \leq \int_0^T \mathrm{d}s\, \mathbb{E}_{\boldsymbol{Z}}[L(\boldsymbol{z};s)], \tag{72}$$

where,

$$L(\boldsymbol{z};s) := \mathbb{E}_{\boldsymbol{X},\tilde{\boldsymbol{X}}_s}\left[\frac{\sigma^2}{2}\left\|\nabla \log p_{\text{eq}}^{(s)}(\tilde{\boldsymbol{x}}_s) - \nabla \log p(\tilde{\boldsymbol{x}}_s, s|\boldsymbol{x}, 0) + \boldsymbol{e}_{\boldsymbol{\theta}}(\tilde{\boldsymbol{x}}_s, s; \boldsymbol{z})\right\|^2\right], \tag{73}$$

$$h(s_1, s_2) := \int_{s_1}^{s_2} \mathrm{d}s\, \mathbb{E}\left[\frac{\sigma^2}{2}\|\nabla \log p(\tilde{\boldsymbol{x}}_s, s|\tilde{\boldsymbol{x}}_{s_1}, s_1)\|^2 + \nabla \cdot b_+\right]. \tag{74}$$

The average in Eq. (74) is taken over all paths that start from $p(\tilde{\boldsymbol{x}}_{s_1}, s_1)$, which is $p_{\text{d}}(\boldsymbol{x})$ forward diffused from $s = 0$ to $s = s_1$. Notice that if we integrate $L$ from an intermediate time $s = \tau$ up to $T$, the l.h.s. in Eq. (72) must be updated with the reconstructed density at $\tau$,

$$\mathbb{E}_{\tilde{\boldsymbol{X}}_\tau, \boldsymbol{Z}_{\text{sem}}}[-\log p_{\boldsymbol{\theta}}(\tilde{\boldsymbol{x}}_\tau, \tau|\boldsymbol{z}_{\text{sem}})] + h(\tau, T) - \mathbb{E}_{\tilde{\boldsymbol{X}}_T}[-\log p_0] \leq \int_\tau^T \mathrm{d}s\, \mathbb{E}_{\boldsymbol{Z}}[L(\boldsymbol{z}_{\text{sem}};s)], \tag{75}$$

where $h(\tau, T)$ and $\mathbb{E}_{p_0}[-\log p_0]$ are still independent of $\boldsymbol{\theta}$. The latent $\boldsymbol{z}_{\text{sem}}$ encodes $\tilde{\boldsymbol{x}}_\tau$, the version of $\boldsymbol{x}$ that has been forward diffused for a time $\tau$. Following our earlier logic, $\boldsymbol{z}_{\text{sem}}$ manages to evade much of the perceptual information—it 'sees' images where most of these microscopic details have been washed out, and only the semantic structure remains—if $\tau$ is chosen judiciously. We can introduce another latent, $\boldsymbol{z}_{\text{per}}$, to aggregate the information from $s \in (0, \tau)$. That is, we run the controlled SDE from $t = T - \tau \to T$, or equivalently $s = \tau \to 0$, in Eq. (68) (see Fig. 19). We assume that the controlled evolution starts at $p_{\boldsymbol{\theta}}(\tilde{\boldsymbol{x}}_\tau, \tau|\boldsymbol{z}_{\text{sem}})$, the state prepared from $p_0$ by the $\boldsymbol{z}_{\text{sem}}$-conditioned drift. Thus,

$$\mathbb{E}_{\tilde{\boldsymbol{X}}_\tau, \boldsymbol{Z}_{\text{per}}}[-\log p_{\boldsymbol{\theta}}(\tilde{\boldsymbol{x}}_0, 0|\boldsymbol{z}_{\text{per}})] + h(0, \tau) - \mathbb{E}_{\tilde{\boldsymbol{X}}_\tau, \boldsymbol{Z}_{\text{sem}}}[-\log p_{\boldsymbol{\theta}}(\tilde{\boldsymbol{x}}_\tau, \tau|\boldsymbol{z}_{\text{sem}})] \leq \int_0^\tau \mathrm{d}s\, \mathbb{E}_{\boldsymbol{Z}}[L(\boldsymbol{z}_{\text{per}};s)], \tag{76}$$

Combining Eqs. (75) and (76), we find that Eq. (72) can be rewritten with a split loss

$$\boxed{\begin{aligned}
&\mathbb{E}_{\boldsymbol{X},\boldsymbol{z}_{\text{sem}},\boldsymbol{z}_{\text{per}}}[-\log p_{\boldsymbol{\theta}}(\boldsymbol{x}|\{\boldsymbol{z}_{\text{sem}}, \boldsymbol{z}_{\text{per}}\})] + h(0,T) - \mathbb{E}_{\tilde{\boldsymbol{X}}_T}[-\log p_0] \\
&\leq \int_0^\tau \mathrm{d}s\, \mathbb{E}_{\boldsymbol{Z}_{\text{per}}}[L(\boldsymbol{z}_{\text{per}};s)] + \int_\tau^T \mathrm{d}s\, \mathbb{E}_{\boldsymbol{Z}_{\text{sem}}}[L(\boldsymbol{z}_{\text{sem}};s)] =: \mathcal{L}_{\text{DEM}}^{\text{split}}.
\end{aligned}} \tag{77}$$

Thus, $\boldsymbol{Z}_{\text{sem}}$ and $\boldsymbol{Z}_{\text{per}}$ access information from different intervals of the forward diffusion process (see Fig. 16). We can verify points 4 and 5 by examining each of these latents closely.

We begin by visualizing $\boldsymbol{Z}_{\text{sem}}$ and $\boldsymbol{Z}_{\text{per}}$ for DAE's trained on MNIST and CIFAR-10 (see Fig. 17). We use $D_{\boldsymbol{Z}} = 20$ for the former and $D_{\boldsymbol{Z}} = 60$ for the latter, for both semantic and perceptual latents. These are generated by separate convolutional encoders. Optionally, we can adjust the receptive field of $\boldsymbol{Z}_{\text{sem}}$ to be larger than that of $\boldsymbol{Z}_{\text{sem}}$ by increasing the number of encoder layers, as we do. These are mapped to two-dimensional space using t-SNE for easier visualization

## 2D t-SNE plot of MNIST latents

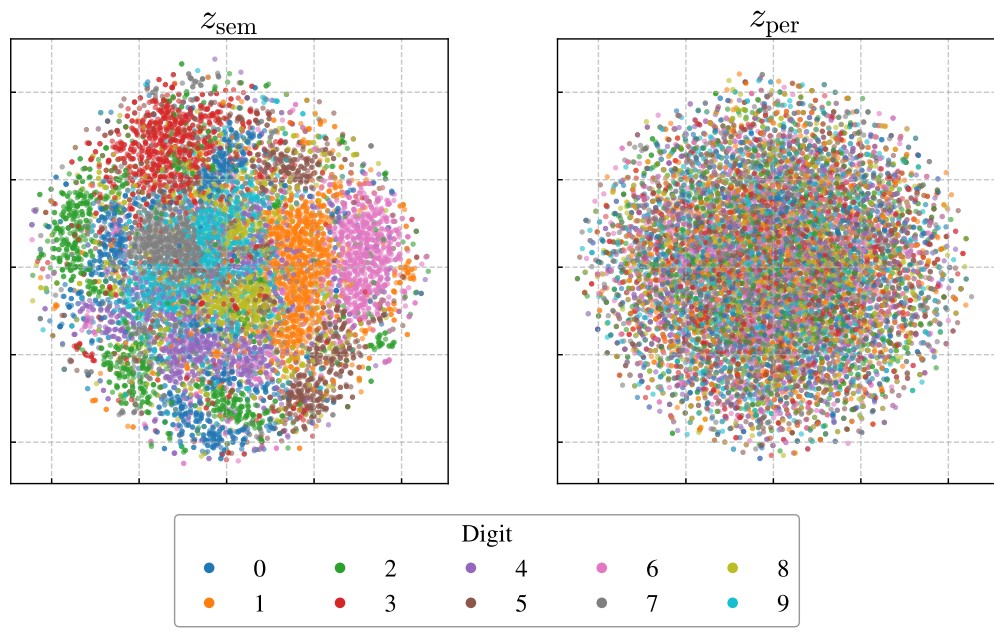

*(a)* A 2D t-SNE plot of the 20-dimensional latents $z_{\mathrm{sem}}$ and $z_{\mathrm{per}}$ produced by a DAE trained on MNIST digits. Information erased by the forward process up to $\tau = 0.1T$ is encoded in perceptual latent $z_{\mathrm{per}}$, whereas all information beyond this point is captured by the semantic latent $z_{\mathrm{sem}}$. Clusters of $z_{\mathrm{sem}}$ correspond to different MNIST digits. On the other hand, $z_{\mathrm{per}}$ shows little structure because the textural details of the images are very evenly distributed amongst all the digit classes.

## 2D t-SNE plot of CIFAR-10 latents

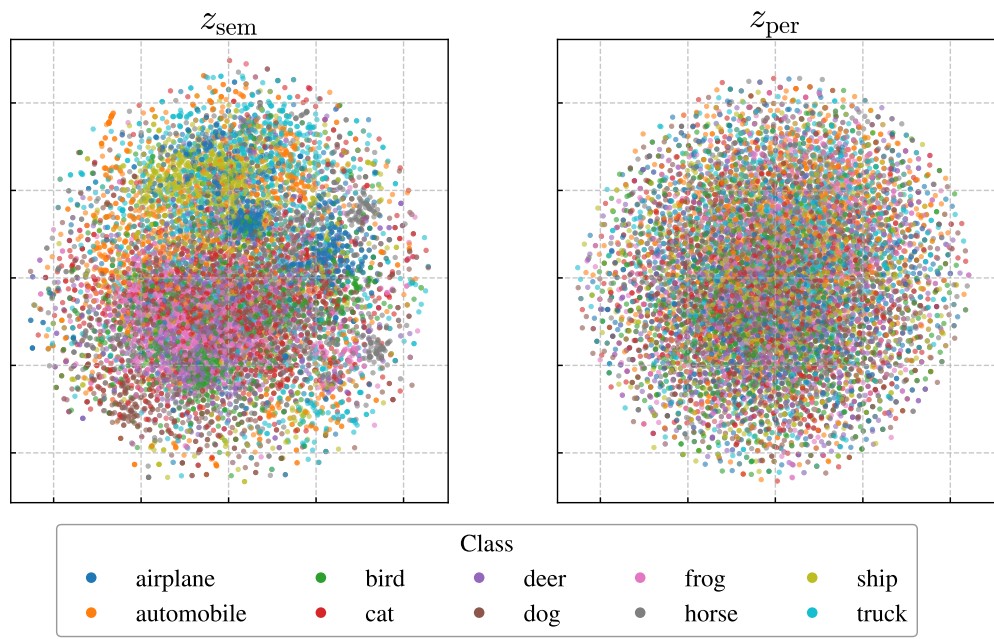

*(b)* t-SNE for CIFAR-10 latents. The more nuanced structure of $z_{\mathrm{sem}}$ reflects the far higher semantic variation between images in CIFAR-10. Both $z_{\mathrm{sem}}$ and $z_{\mathrm{per}}$ had $D_{Z} = 60$ dimensions.

*Figure 17*

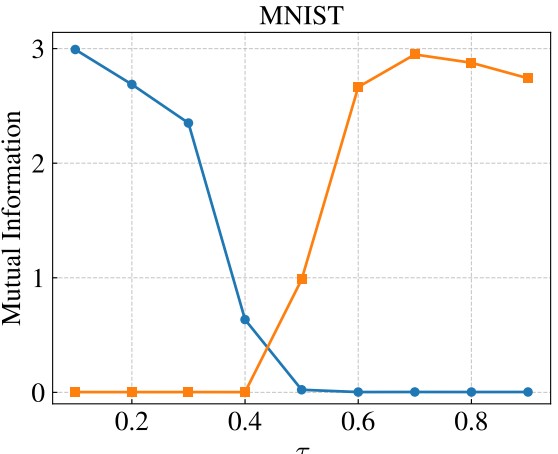 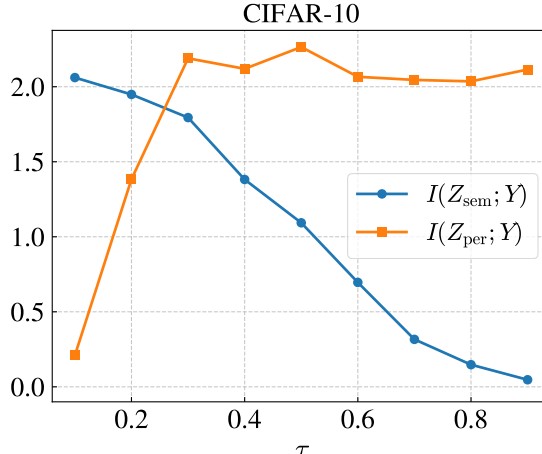

*Figure 18.* Mutual information between the image labels $Y$ and the corresponding semantic and perceptual latents, as a function of the partitioning time $\tau$ (cf. Eq. (77)). At early $\tau$, the semantic latent is strongly correlated with the labels, whereas the perceptual details are completely independent. As $\tau$ increases, $Z_{\text{sem}}$ becomes progressively irrelevant, whereas $Z_{\text{per}}$ does the opposite—knowing enough small-to-medium details helps the model understand what the image is.

(van der Maaten & Hinton, 2008). With $\tau = 0.1T$, we find that there is little to no structure in $Z_{\text{per}}$ in either case, in agreement with our claim that images from different classes have similar small-scale details. On the other hand, clusters of $Z_{\text{sem}}$ appear in the t-SNE plot, showing that class labels correspond to large-scale features robust to small perturbations.[5]

We can do better than inspect $Z_{\text{sem}}$ and $Z_{\text{per}}$ by eye. Recall that Eq. (12) can be used to estimate the mutual information between random variables. By training a small diffusion model on pairs $\{(z_{\bullet}, y)\}_{i=1}^{N}$, we can determine $I(Z_{\bullet}, Y)$ approximately. The results are plotted in Fig. 18 for a range of $\tau$ values. As expected, $Z_{\text{sem}}$ is correlates well with $Y$ at small $\tau$, whereas $Z_{\text{per}}$ is nearly independent of it. It should be pointed out that $I(Z_{\bullet}; Y) \leq I(X; Y)$, so there *can* be class-specific information in the noised image in $(0, \tau]$ that is not encoded in $Z_{\text{per}}$ for small $\tau$. However, as $\tau$ is increased, $Z_{\text{per}}$ rapidly encodes class information. We speculate that the $X \to Z_{\text{per}}$ encoder can detect semantic meaning if it is given sufficient information about the medium-scale features, since an image is the sum of its parts. Furthermore, if $\tau$ is not too close to $s = 0$, the encoder focuses effort on information that differentiates the images, and downplays the shared textural detail between them. This is why Fig. 15 showed *some* clustering in the DAE case—even with the large amount of small-scale information the diffusion decoder captures, the encoder is incentivized to construct latents that uniquely identify the images (cf. Sec. E.1). We also mention in passing that the cross-over phenomenon in Fig. 15 is reminiscent of the *critical windows* of feature emergence (Li & Chen, 2024).

**Hyperparameter summary**   Table 1 lists the hyperparameters used for the experiments in Fig. 16. The diffusion model uses a U-net with self-attention based on the architecture from (Salimans & Ho, 2022); the hyperparameters in Table 1 follow the conventions in that paper. This is also the same network used for the diffusion decoder in the DAE experiments in Figs. 17 and 18, with the class embedding replaced with a latent embedding layer. The experiments with the joint Gaussian in Sec. D.1 used a diffusion model with a simple MLP backbone. Two separate models were trained to approximate the conditional and marginal scores from the samples. These scores were used in the MINDE expression, Eq. (12), to produce a neural estimate of $I(X; Y)$ in Figs. 8 and 13. The same approach is also used to estimate $I(Z_{\bullet}; Y)$ in Fig. 18.

All experiments use the VP SDE of (Song et al., 2021b) with a linear schedule $\beta(s) = \beta_{\min} + (\beta_{\max} - \beta_{\min})s$ integrated over $s \in [10^{-5}, 1]$, where $\beta_{\min} = 0.1$ and $\beta_{\min} = 16$. Models are trained with the Adam optimizer (Kingma & Ba, 2014). Neural entropies and mutual information are estimated by Monte Carlo on a test batch of 500 samples from the test dataset, with the entropy rate evaluated on a uniform grid of 100 points in $s \in [10^{-5}, 1]$. For images, the unconditional neural entropy $S_{\text{NN}}^{X}$ is computed by passing the conditional model a null class label, so that the same network is used for both conditional and marginal estimates.

---

[5]This is why the diffusion classifiers from (Clark & Jaini, 2023; Li et al., 2023) employ a denoising objective that significantly downweights the contributions from the earlier time steps. The popular practice of 'variance-dropping' also achieves a similar effect (see Sec. B.3).

*Table 1.* Hyperparameters for the experiments on MNIST, CIFAR-10, and Tiny ImageNet from Fig. 16.

| Setting | MNIST | CIFAR-10 | Tiny ImageNet |
|---|---|---|---|
| *Data* | | | |
| Image shape | $28 \times 28 \times 1$ | $32 \times 32 \times 3$ | $64 \times 64 \times 3$ |
| Training set size | 60,000 | 50,000 | 100,000 |
| *U-Net architecture* | | | |
| Base channels | 256 | 256 | 128 |
| Embedding channels | 1024 | 1024 | 512 |
| Channel multipliers | $[1, 1, 1]$ | $[1, 1, 1]$ | $[1, 2, 2, 4]$ |
| Residual blocks per level | 1 | 2 | 2 |
| Attention resolutions | $\{7, 14\}$ | $\{8, 16\}$ | $\{16, 32\}$ |
| Attention heads | 1 | 1 | 4 |
| Dropout | 0.2 | 0.2 | 0.1 |
| *Encoder (for DAE)* | | | |
| Conv features | $[32, 64, 128]$ | $[32, 64, 128]$ | N/A |
| *Training* | | | |
| Batch size | 32 | 32 | 256 |
| Learning rate | $1 \times 10^{-4}$ | $2 \times 10^{-4}$ | $2 \times 10^{-4}$ |
| Epochs | 50 | 50 | 100 |

# F. Notation

The natural logarithm is denoted by $\log$. We use the symbol $S := -\int p \log p$ for differential entropy, and the mutual information $I(X; Y)$ is in nats. Scalars are written in plain letters, while boldface symbols such as $\boldsymbol{X}, \boldsymbol{Y}, \boldsymbol{Z}$ denote higher-dimensional random variables. We write $\boldsymbol{x}$ for a realization of $\boldsymbol{X}$, with unsubscripted symbols always referring to the data distribution $p_{\mathrm{d}}$. We also write $p_{\mathrm{d}}(\boldsymbol{x}, \boldsymbol{y})$ for the joint data distribution.

We use the time variable $s$ for the forward diffusion process, which runs from left ($s = 0$) to right ($s = T$) in Fig. 19. $\hat{\boldsymbol{B}}_s$ and $\boldsymbol{B}_t$ denote the Brownian motions associated with the forward and reverse/controlled SDEs, respectively. $\nabla$ is the gradient with respect the spatial coordinates, and $\partial_t, \partial_s$ are partial time derivatives. $S_{\mathrm{tot}}$ is the total entropy produced during forward diffusion, and is closely approximated by the neural entropy $S_{\mathrm{NN}}$. The time-dependence of the entropies is implicit in most of the main text; $S_{\mathrm{tot}}$ and $S_{\mathrm{NN}}$ without the time argument should be understood as $S_{\mathrm{tot}}(s = T) \equiv S_{\mathrm{tot}}(T)$.

The density $p(\tilde{\boldsymbol{x}}_s, s)$ is the same as $p(\boldsymbol{x}_t, t)$. That is, the symbol $p$ is overloaded so we do not have to write $p(\cdot, s) = p(\cdot, T - t)$ everywhere. Throughout the paper, we set Boltzmann's constant to unity, $k_{\mathrm{B}} = 1$. $p_{\mathrm{d}}$ and $p_0$ denote the initial ($s = 0$) and final ($s = T$) densities for the forward process, and $p_{\mathrm{eq}}$ is its equilibrium state. Diffusion takes an infinite time to equilibrate, but we always take $T$ to be large compared to the intrinsic time scale of the diffusion process, so $p_0 \approx p_{\mathrm{eq}}$.

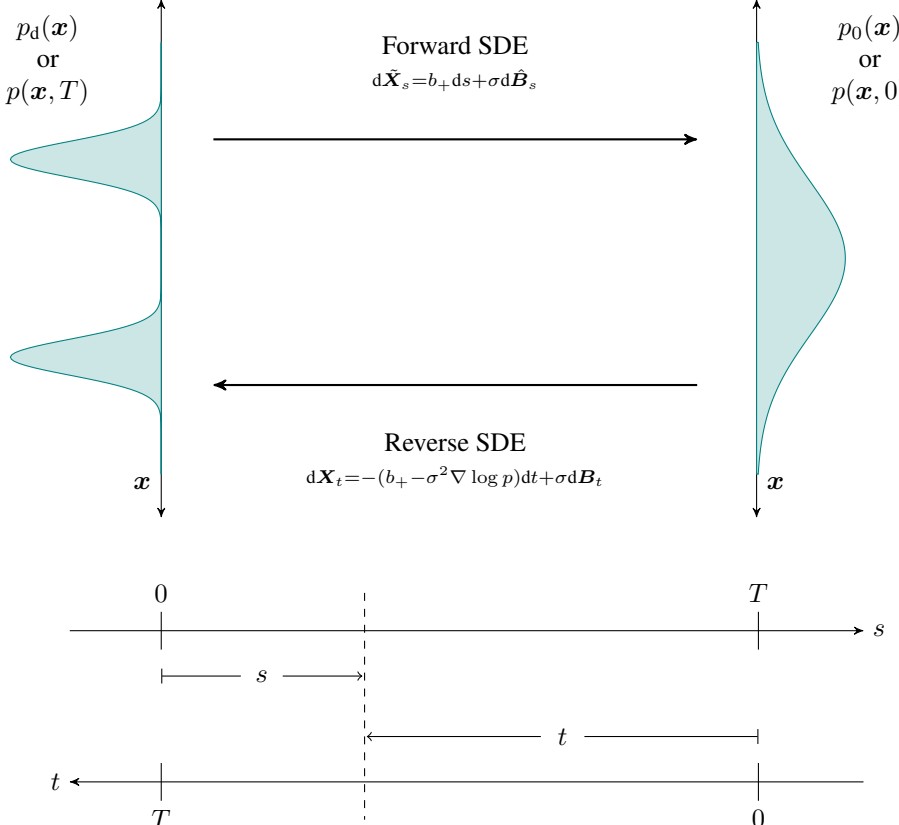

*Figure 19.* A schematic of the forward and reverse diffusion processes.

