# OpenReview forum: "On the Separability of Information in Diffusion Models"
_ICML.cc/2026/Conference — ICML 2026 regular_

### Official Review · Reviewer_EF2a · 2026-02-23

**Soundness:** 3
**Presentation:** 2
**Significance:** 3
**Originality:** 3
**Overall Recommendation:** 4
**Confidence:** 3

**Summary:**

The paper analyzes how information is “stored” and used in pixel-space diffusion models via the neural entropy framework, decomposing the overall information budget into within-image dependencies (via total correlation) and conditioning information (via mutual information). It argues that, for images, a large portion of neural entropy is devoted to reconstructing fine perceptual details and manifold localization, while class-related information is mostly semantic and appears earlier in the reverse process. The paper further connects this separation to classifier-free guidance (CFG), claiming guidance amplifies mutual information early but naturally tapers as perceptual details are filled in. Empirical evidence is provided via (i) a joint Gaussian toy model with analytical scores and (ii) diffusion-autoencoder probes that split “perceptual” vs “semantic” latents over diffusion time.

**Compliance With Llm Reviewing Policy:**

Affirmed.

**Key Questions For Authors:**

Please refer to the weaknesses.

**Limitations:**

yes

**Strengths And Weaknesses:**

# Strengths
* Clear conceptual decomposition of “information in diffusion” into (a) information to match the marginal distribution and (b) extra information for conditioning, with an explicit connection to mutual information via Eq. (10) and the score-difference estimator in Eq. (12)–(14).
* The link between the mutual-information estimator and the CFG “classification vector” (Eq. (14) and discussion around Eq. (15)) is a useful unifying perspective that helps interpret why CFG tends to affect global structure more than textures.
* The diffusion-autoencoder probing setup (Eq. (16)–(17), Figure 5) is a reasonably direct attempt to test the paper’s semantic-vs-perceptual claim without relying on ground-truth scores for images.
---
# Weaknesses
* Lack of tables and concrete numeric reporting is a real problem for an ICML paper. There are no results tables summarizing experimental settings or outcomes, even though the paper makes quantitative claims like Fig.3, Fig.4, and Fig.5. A reader cannot easily verify, compare, or reuse these findings.
* The Gaussian experiments are conceptually sound as a controlled testbed, but the image experiments depend on a custom DAE split and mutual-information estimation procedure with several moving parts. The main paper does not provide enough implementation/training detail (architectures, compute, seeds, optimizer settings) to fully judge robustness.
* Image-domain CFG explanation is not empirically validated in the main paper. The strongest CFG evidence is Gaussian (Figure 2). The paper does not show, for actual image diffusion, that the guidance vector norm tapers as textures are filled in, despite presenting this as an explanation of CFG efficacy (Abstract, Section 3, Section 4).
* Generalization beyond small datasets is unproven. MNIST/CIFAR-10 are informative but limited. The paper’s claims seem intended to explain behavior of modern conditional diffusion at scale; the main-text evidence does not meet that bar.

---

> ### Author Rebuttal · Authors · 2026-03-29
>
> Thank you for your review. We address your comments below.
>
> 1. We will include tables in the revision to consolidate the information you requested.
>
> 2. The details of the DAE probe experiments will be added in the revised version. At present this information is largely present in the supplementary material (code).
>
> 3. You are touching on a very important point. One of the first things we tried was to plot $\Vert e_{\theta}(x_s, s; y) - e_{\theta}(x_s, s) \Vert^2$ over time $s$ for the image experiments, just like we did in Fig. 2 for the joint Gaussian model (the green curve in the leftmost plot). However, the very high dimensionality of the image data (order $D_X \approx 10^3$ for images vs. order $D_X = 25$ for the Gaussian) make both $e_{\theta}(x_s, s; y)$ and $e_{\theta}(x_s, s)$ rise much sharply, and to much larger values, near $s = 0$. Consequently, it is difficult to reliably resolve the equivalent of the green curve in the image case: it would be a very sharp spike under the dashed curves in Fig. 3. This was one of the main reasons we resorted to the DAE for our analysis.
>
>    By the arguments in Sec. 3, the classification vector $s_{\rm cl}(y, s) = e_{\theta}(x_s, s; y) - e_{\theta}(x_s, s)$ will be largest in those time intervals that source most of the value of $I(X;Y)$. The latents $Z_{\rm per}$ and $Z_{\rm sem}$ encode the _cumulative_ information lost from $X$ in the intervals $(0, \tau]$ and $(\tau, T]$ respectively. Thus, by measuring $I(Z_{\rm per}; Y)$ and $I(Z_{\rm sem}; Y)$ we are looking at two quantities that integrate the spike, and more importantly, we are able to _compare_ information from different intervals explicitly. A small value of $I(Z_{\rm per}; Y)$ at early $\tau$ tells us that the diffusion model sees little class-specific information there, and by extension the contribution to CFG from this interval is negligible.
>
> 4. Reviewer GRFd also raised this concern, and we have responded to it in our rebuttal there. Briefly, the separability argument becomes _stronger_ at higher image resolutions due to the difference in how ${\rm TC}(X)$ and $I(X;Y)$ scale. We already see this trend in the experiments with MNIST, CIFAR-10, and Tiny ImageNet.
>
> We hope this allays your concerns. We will incorporate the presentation changes you mentioned into the revised version.

---

> > ### Author Rebuttal · Reviewer_EF2a · 2026-04-01
> >
> > The authors’ rebuttal has addressed my concerns, and I have decided to raise my rating to Weak Accept.

---

### Official Review · Reviewer_GRFd · 2026-03-12

**Soundness:** 3
**Presentation:** 3
**Significance:** 3
**Originality:** 3
**Overall Recommendation:** 5
**Confidence:** 4

**Summary:**

This paper provides an information-theoretic analysis of diffusion models, demonstrating that the information a model stores (neural entropy) can be decomposed into two distinct parts: Total Correlation (TC), which is the large cost of reconstructing low-level perceptual details, and Mutual Information (I(X;Y)), the smaller amount needed to link data with a semantic condition. It shows that in image generation, these two types of information are temporally separated—semantics are determined early in the process, while fine details are added later. This framework explains why Classifier-Free Guidance (CFG)​ works: it amplifies the semantic, conditional information (I(X;Y)) primarily in the early stages. The claims are supported by theory, a Gaussian toy model, and image-based autoencoder experiments.

**Compliance With Llm Reviewing Policy:**

Affirmed.

**Final Justification:**

My concerns have been addressed. I'll raise my scores to Accept.

**Key Questions For Authors:**

1. Does the clear separation between "semantic" and "perceptual" information, and the extreme ratio, which were validated on lower-resolution datasets, still hold for high-resolution, semantically complex natural images (e.g., ImageNet 256x256)?

2. To what extent does the authors believe this theoretical framework can generalize to scenarios like text-to-image generation, where the conditional signal Y (text) is itself high-dimensional and semantically rich?

**Limitations:**

The authors have discussed the potential negative societal impact of their work.

**Strengths And Weaknesses:**

Strengths:

Building on prior work, the paper's main novelty lies in its explicit decomposition of a model's information into Total Correlation (for internal, perceptual details) and Mutual Information (for external, semantic conditioning), demonstrating their extreme imbalance and temporal separation. The narrative is coherent, the technical foundation is rigorous, and the experimental design is clever and supportive.

Weaknesses:

The primary weakness lies in the scope of the experiments. As noted, the validation relies mainly on classic, low-resolution datasets (MNIST, CIFAR-10, Tiny ImageNet-64). While the theoretical analysis is general, its applicability to more complex, high-resolution data (e.g., ImageNet 256x256, text-to-image models) remains less thoroughly explored. The hierarchical nature of perceptual details and the complexity of semantic structures in high-resolution images may pose new challenges.

---

> ### Author Rebuttal · Authors · 2026-03-28
>
> Thank you for the review. We address your concerns below.
>
> **Questions:**
>
> 1. As image resolution increases the pixel count grows quadratically, whereas the intrinsic dimensionality of the data manifold grows much more slowly. There are far more nearest-neighbor pixel correlations to resolve, so ${\rm TC}(X)$ explodes, while $I(X;Y) \leq H(Y)$ grows logarithmically in the number of classes. Thus, the ratio $I(X;Y)/S_{\rm NN}^{X}$ gets _worse_, which strengthens the separability argument. We already see this in Fig. 16, where $S_{\rm NN}^{X}$ for Tiny ImageNet is an order of magnitude larger than that of MNIST, but $I(X;Y)$ merely doubles.
>
> 2. For text prompts $Y$ carries more information than a simple class label, so $H(Y)$ is larger, but the fundamental asymmetry between $S_{\rm NN}^{X}$ and $I(X;Y)$ would still persist since $H(Y)$ remains small compared to the information cost of resolving the full pixel-space manifold, which grows rapidly with image resolution. Furthermore, as we point out in the conclusion, the success of latent diffusion models is evidence for our thesis at scale, since this architecture is advantaged by offloading the perceptual reconstruction to a separate network.
>
> **Weaknesses:**
>
> Your concern about the experimental scope is understandable. We wanted to include more experiments with larger datasets, however, these experiments become very expensive for us at that scale. This is especially true because the diffusion autoencoder probe we developed requires that (1) we train the setup from scratch, since standard pretrained weights do not follow this framework, and (2) we need to train a new DAE for _each_ value of the separation scale $\tau$.
>
> That said, the scaling arguments above, together with the empirical trend from the three datasets MNIST $\to$ CIFAR-10 $\to$ Tiny ImageNet, demonstrate that the separability of semantic/perceptual information becomes _stronger_ at higher resolutions. We can update the revision to make this more explicit. We will try to include additional experiments if we can find the resources to do so.
>
> Thank you once again for the review. We hope the discussion clarifies the generality of our claims.

---

> > ### Author Rebuttal · Reviewer_GRFd · 2026-04-02
> >
> > My concerns have been addressed.

---

### Official Review · Reviewer_Paid · 2026-03-14

**Soundness:** 3
**Presentation:** 2
**Significance:** 3
**Originality:** 3
**Overall Recommendation:** 5
**Confidence:** 3

**Summary:**

In this paper, the authors aim to understand diffusion models and CFG from a information centric perspective. The authors propose neural entropy $S_\text{tot}^X$ of a generated sample $X$, which is basically the squared norm of predicted noise across sampling steps, and larger entropy means more **effort/information budget** for generating a sample $X$.

Then they extend the concept to conditional generation, which induces $S_\text{tot}^{X|Y}$, the entropy for generating $X$ conditioning on $Y$. They also propose $I(X, Y)=S_\text{tot}^{X|Y}-S_\text{tot}^{X}$ which is the additional information required to narrow down sampling to a more specific conditional distribution $p(x|y)$, and $I(X, Y)$ can be correlated with the difference of the conditional score and unconditional score.

They further discuss the meaning and implications of $I(X, Y)$: They show that larger guidance strength improves $I(X,Y)$, meaning there will be a stronger correlation between $X$ and $Y$. They also study the learnability, by comparing $I(X,Y)$ with $S_\text{tot}^{X}$ on a controlled gaussian data, and they found that learning $I(X,Y)$ can be easy and irrelevant to learning data manifold or $S_\text{tot}^{X}$, accounting the success of conditional training and CFG. They also conducted empirical studies on CIFAR10 and MNIST.

**Compliance With Llm Reviewing Policy:**

Affirmed.

**Final Justification:**

Please see **Acknowledgement**. The framework is interesting, but missing some novel observations currently.

**Key Questions For Authors:**

1. In Fig. 2. The authors show that $I(X, Y)$ can saturate when we increase CFG scale $w$. However, when we keep increasing $w$ in practice, we get the distorted or oversaturated samples. It seems neural entropy can not explain such a defect?
2. In many parts of the paper, the authors claim that $I(X, Y)$ is stored *on top of* $S_\text{tot}^{X}$. That is not necessarily true, because we may be training separate networks for conditional and unconditional scores. For instance, the famous EDM [1] framework does so.
3. Recent work [2] that studies how a ReLU-MLP learns under diffusion loss can be a good complement to the neural entropy perspective. It shows networks can either store locally sparse samples in their weights, or learn statistics/PCs and form a compact representation to approximate the underlying distribution. The observation that networks build compressive internal representations may justify how networks efficiently absorb the information from the training set.


[1] Karras, Tero, et al. *"Elucidating the design space of diffusion-based generative models."* NeurIPS 2022.

[2] Zhang, Zekai, et al. *"Generalization of Diffusion Models Arises with a Balanced Representation Space."* ICLR 2026.

**Limitations:**

Please see **Weakness**.

**Strengths And Weaknesses:**

### Strength:

1. The information-centric view of how diffusion and CFG work is charming. The controlled study on learning $I(X,Y)$ may provide a justification for why conditional training is effective, which is novel.
2. The authors introduce and discuss the concepts rigorously; the arguments seem self-contained and grounded. The paper can be overwhelming at first glance, but it gets interesting when looked at carefully.



### Weakness:

1. It seems we cannot efficiently estimate $I(X,Y)$ from a given pretrained model due to the imperfect training. And I feel like that's one reason this paper falls a bit short on application and predictive claims.

2. Following point 1, to estimate $I(X,Y)$, the authors would need to train several additional diffusion autoencoders for dimension reduction, and estimate the mutual information on the learned represenations $Z(X)$. This is not standard or scalable; and the training of diffusion autoencoder can still lead to imperfection. I feel like this part is particularly confusing.

---

> ### Author Rebuttal · Authors · 2026-03-28
>
> Thank you for the positive comments and many interesting questions. We address them below.
>
> **Weaknesses:**
>
> 1. You are correct in that the MINDE estimator can produce biased values of $I(X;Y)$ when a pertained model is used. This is studied rigorously in Franzese et. al. (2024). In this paper, our goal is to demonstrate the separability of the semantic/perfecptual information, and this conclusion is unaffected by the imperfections in the MINDE estimator, as we discuss in point 2 in the rebuttal to reviewer jYnY. In that sense the paper advances our understanding of the theory of diffusion models. However, we are presently working on an application of these insights to diffusion distillation, and we hope to have this work published soon.
>
> 2. There may be some confusion here, so please correct us if we did not understand your concern. To estimate $I(X;Y)$ we do not need to use diffusion autoencoders at all; a single diffusion model that approximates $\nabla \log p(x_t, t|y)$ and $\nabla \log p(x_t, t)$ (or two models, one for each of them, per your later comment) is sufficient to compute $I(X;Y)$ with the MINDE formula. The later experiments with the DAE is to prove that information erased in the early stages of the forward diffusion process carry little information relevant to predicting the class label.
>
> **Questions:**
>
> 1. This is a good point. As shown in the plots on the left in Fig. 2, if $w$ is very large CFG can produce samples that are dramatically different from those of the true data distribution. We believe this is also what happens in case of images, where strong $w$'s produce distorted samples as you described.
>
> 2. We should clarify what we mean when we say $I(X;Y)$ is stored on top of $S_{\rm tot}^{X}$.
> This is just the statement that $S_{\rm tot}^{X|Y} = S_{\rm tot}^{X} + I(X;Y)$. That is, a conditional diffusion model that is trained to produce $X$ conditioned on $Y$ must store an additional amount of information compared to a model that generates $X$ unconditionally. As you point out, it is possible to train a single network to model the $\nabla \log p(x_t, t|y)$'s as well as $\nabla \log p(x_t, t)$ OR use two separate networks for these. In the latter case the unconditional network contains $S_{\rm NN}^{X} \approx S_{\rm tot}^{X}$ worth of information, whereas the conditional one contains $S_{\rm NN}^{X|Y} \approx S_{\rm tot}^{X|Y}$, which exceeds the other one by $I(X;Y)$. This is what we meant.
>
> 3. Thank you for the reference. Your point about compressive internal representations is well-taken. We have been thinking along a similar direction recently, inspired by the Kamb and Ganguli paper from 2025. Roughly, our intuition is that, by breaking down images to patches diffusion models are constructing a sort of 'basis,' the elements of which can be combined in different ways to produce a wide variety of images that conform with the training data statistics. It would be interesting to combine this with your suggestion about compact representations.
>
> Once again, thank you for the review. Please let us know if we addressed your concerns satisfactorily. We hope it improves your assessment of our work.

---

> > ### Author Rebuttal · Reviewer_Paid · 2026-04-02
> >
> > I thank the authors for the rebuttal. I now understand that with the diffusion autoencoder experiments, the authors are trying to separate the information $Z_\text{sem}$ and $Z_\text{per}$ within $X$ learned at different steps, rather than just to better compute the mutual information, which can be done directly by the MINDE formula. Then the authors try to show $Z_\text{sem}$ forms separable clusters and serves as semantics. But the tSNE separation in Fig.5 is a bit vague (from my experience, internal representations learned within the score network can be much more separable [1] [2]), so though the arguments make sense (and also correlate to previous findings [3] [4]), the presentation is not entirely intuitive from my view.
> >
> > Overall, I still feel like the paper lacks a few new observations, and the authors are mainly justifying the neural entropy framework for CFG. But I am glad to hear there is ongoing progress aiming for more practical applications under this view, and I would love to see that. I have increased my ratings accordingly.
> >
> >
> >
> > [1] Xiang, Weilai, et al. *"Denoising diffusion autoencoders are unified self-supervised learners."* ICCV 2023.
> >
> > [2] Chen, Xinlei, et al. *"Deconstructing denoising diffusion models for self-supervised learning."* ICLR 2025.
> >
> > [3] Wang, Binxu, and John J. Vastola. *"Diffusion models generate images like painters: an analytical theory of outline first, details later."* TMLR 2025.
> >
> > [4] Li, Xiao, et al. *"Understanding representation dynamics of diffusion models via low-dimensional modeling."* NeurIPS 2025.

---

> > > ### Author Response · Authors · 2026-04-03
> > >
> > > Thank you for the discussion, and the revised score.
> > >
> > > Re: the internal representation within the score network, the point of our DAE experiments is to treat the network *as a whole* rather than examining its parts. This is the same philosophy as the neural entropy paper. They ask "what measurements can we make if we treat the NN as a black-box." In this spirit, $Z_{\rm per}$ and $Z_{\rm sem}$ aggregate and compress the information the entirety of the network at different points of time. The point of these experiments is not to create the cleanest cluster separation as much as it is to show how much class-specific information is stored at different points of time.
> > >
> > > Thank you for the references. We had seen the work by Xiang before. You may also find [1] interesting. We are excited for you to see our follow-up work.
> > >
> > > [1] Kadkhodaie, Z., Mallat, S., & Simoncelli, E. P. (2025). Unconditional CNN denoisers contain sparse semantic representations of images.

---

### Official Review · Reviewer_jYnY · 2026-03-20

**Soundness:** 3
**Presentation:** 3
**Significance:** 3
**Originality:** 3
**Overall Recommendation:** 5
**Confidence:** 3

**Summary:**

This paper provides an information-theoretic framework to explain how conditional diffusion models allocate capacity. It mathematically and empirically demonstrates that most of the neural entropy is used on resolving the low-dimensional data manifold (perceptual details), while semantic aspects require comparatively little information and semantic conditioning occurs early in generation.

**Compliance With Llm Reviewing Policy:**

Affirmed.

**Key Questions For Authors:**

See weaknesses

**Limitations:**

yes

**Strengths And Weaknesses:**

### **Strengths

* Elegant and mathematically grounded unification.
* Deep, first-principles geometric explanation for why CFG works well.
* Clever and effective experiments to separate and measure semantic versus perceptual latents.

### Weaknesses

* The authors equate differential entropy with "information stored in the neural network." Maybe I'm taking this too literally, but it feels a bit imprecise, as I think the algorithmic cost to collapse a distribution does not necessarily map to network memory capacity.
* The MINDE estimate grossly exceeds the theoretical upper bounds. The authors acknowledge this as a "ballpark estimate," but this seems like a very large structural bias that might make the quantitative comparisons a bit fragile.
* For the CFG saturation analysis, the authors trained new models to estimate MI on a toy joint gaussian model. I could be missing something, but doesn't linear CFG preserve gaussianity, and therefore couldn't this have been calculated analytically?
* About the attribution of the large entropy cost to TC: isn't TC basis-dependent? E.g. what happens to TC if we rotate the data into an independent basis like PCA? Maybe it's safer to attribute this entropy divergence purely to the low intrinsic dimensionality of the manifold?

---

> ### Author Rebuttal · Authors · 2026-03-27
>
> Thank you for the kind comments, and the genuinely thoughtful feedback. We address them below.
>
> 1. You are correct in that $S_{\rm NN}$ does not map directly to the total bits in all the parameters of the underlying neural network. But it is still the correct measure of information for diffusion models, as explained in Premkumar (2025). Here is the core argument: when we say that a standard ASCII character is 7 bits, what we mean is that we require $-\log_2 p = -\log \frac{1}{128} = 7 \ \textrm{bits}$ to locate one specific character from 128 possibilities. In the same way, $S_{\rm NN} \approx -\frac{1}{N} \log \mathcal{P}[p_{\rm d}]$ measures (approximately) the amount of information needed to locate (nearly) the data distribution $p_{\rm d}$ from the family of distributions that $p_{\rm eq}$ can fluctuate to. In other words a diffusion model "stores an entire distribution," and not $N$ individual training samples. That said, we do suspect there is some connection between $S_{\rm NN}$ and the parameter bit count, which we hope to uncover at some point.
>
> 2. We agree that the MINDE estimate is susceptible to the sort of error you see in our experiments. Theoretical guarantees about MINDE accuracy are discussed in detail in Sec. 3.1 of Franzese et. al. (2024). The focus of our paper was to demonstrate that $I(X;Y)$ is sourced from a different part of the diffusion process than the one which resolves microscopic detail. In this context, what is important is the ratio of $I(X;Y)$ to $S^X_{\rm NN}$ which is $\approx 10^{-4} \text{ to } 10^{-3}$ in our experiments. The MINDE errors do little to affect this ratio. Furthermore, in the DAE probe experiments, we are more concerned with the _trends_ in $I(Z_{\bullet}; Y)$ rather than their absolute values, therefore the MINDE estimates are still meaningful despite their biases.
>
> 3. You're right! We missed this point entirely. We will compute the analytic expressions for $I(X;Y)_{\rm CFG}$ and incorporate it into the plots and appendix. Thank you!
>
> 4. This is a subtle but important point. It is easier to explain with mutual information, before extending the rationale to higher dimensions for ${\rm TC}(X)$. As you intuited, $I(X;Y)$ must be invariant under some diffeomorphic reparameterization. Indeed, it can be shown that $I(f(X); g(Y)) = I(X; Y)$, where $f, g$ are invertible smooth maps. See Theorem 3.7 (f) of *Information Theory: From Coding to Learning* by Polyanskiy and Wu. However, it is _not_ true in general that $I(u(X,Y); v(X,Y))$ is the same as $I(X;Y)$. Your PCA/rotation intuition is an example of this. Intuitively it is because $u(X,Y)$ and $v(X,Y)$ can combine the randomness in $X$ and $Y$ in ways that $f(X)$ and $g(Y)$ cannot; in the rotation case you could create new variables $u$ and $v$ that are not correlated at all, because you mixed $X$ and $Y$ in a specific way such that the randomness in $u$ and $v$ are 'orthogonal.'
>
> We hope this answers your concerns, and improves your appraisal of our work. Thank you again for the great points.

---

> > ### Author Rebuttal · Reviewer_jYnY · 2026-04-05
> >
> > Thanks for the clarifications! I'm not sure I get your point for question 4 though. Do you mean that while TC(X) is not invariant under transformations that mix variables, the chosen coordinate system (e.g. pixel space) is itself semantically meaningful, so TC(X) in this basis is the relevant quantity?

---

> > > ### Author Response · Authors · 2026-04-05
> > >
> > > Indeed. As a simple example, consider a data distribution $p_{\rm d}$ that lives on a $d$-dimensional hyperplane, which itself is embedded in a $D$-dimensional ambient space ($D > d$). If we work in the $d$-component co-ordinate system of the hyperplane, we would see no divergence in the total correlation computed in those co-ordinates. But this is not true of the $D$-component co-ordinates. In the context of diffusion models, squeezing a $D$-dimensional Gaussian onto the hyperplane would require a far greater (infinite in theory) amount of information than that required by a model which transforms a $d$-dimensional Gaussian on the hyperplane to $p_{\rm d}$. The latter does not have to concentrate the probability mass as acutely as the former.

---

### Decision · Program_Chairs · 2026-04-30

**Decision:**

Accept (regular)

**Comment:**

The paper presents a clear and original information-theoretic perspective on conditional diffusion models, arguing for a separation between semantic information relevant to conditioning and the much larger information budget required to reconstruct perceptual detail. Reviewers found the core framework technically interesting, well motivated, and potentially valuable for understanding classifier-free guidance and the internal organization of information in diffusion models. The main concerns focused on experimental scope, presentation, and the reliance on indirect estimation procedures, especially in the image-domain analysis. However, the rebuttal addressed these issues and several reviewers explicitly stated that their concerns were resolved and updated or affirmed their positive scores. Overall, while the empirical validation remains somewhat limited relative to the breadth of the paper’s claims, the contribution is novel, technically solid, and likely to stimulate useful follow-up work in the theory and analysis of diffusion models.